# No More DeLuLu: Physics-Inspired Kernel Networks for Geometrically-Grounded Neural Computation

## Abstract

We introduce the $\mathbb{E}$-product, a kernel operator combining quadratic alignment with inverse-square proximity. We prove it is a Mercer kernel, analytic, Lipschitz on bounded domains, and self-regularizing, admitting a unique RKHS embedding. Neural Matter Networks (NMNs) use $\mathbb{E}$-product as the sole nonlinearity, replacing conventional linear-activation-normalization blocks with a single geometrically-grounded operation. This architectural simplification preserves universal approximation while shifting normalization into the kernel itself via the denominator, rather than relying on separate normalization layers. Empirically, NMN-based classifiers match linear baselines on MNIST while exhibiting bounded prototype evolution and superposition robustness. In language modeling, Aether-GPT2 achieves lower validation loss than GPT-2 with a comparable parameter budget while using $\mathbb{E}$-based attention and MLP blocks. Our framework unifies kernel learning, gradient stability, and information geometry, establishing NMNs as a principled alternative to conventional neural architectures.

## 1. Introduction

Modern neural networks separate geometry from non-linearity: dot products compute alignment, then activation functions like ReLU threshold the result (Goodfellow et al., 2016). This discards information—all negative activations become zero—requiring normalization layers and attention mechanisms to recover expressiveness (Ioffe & Szegedy, 2015; Vaswani et al.,

2017).

We propose the $\mathbb{E}$-product, a neural operator that unifies alignment and proximity in a single computation:

$$\mathbb{E}(\mathbf{w}, \mathbf{x}) := \frac{\langle \mathbf{w}, \mathbf{x} \rangle^2}{\|\mathbf{w} - \mathbf{x}\|^2 + \varepsilon} \qquad (1)$$

Inspired by inverse-square laws in physics, this operator creates a "potential well" around the weight vector $\mathbf{w}$: responses are high when inputs are both aligned *and* close, providing intrinsic non-linearity without thresholding. The $\mathbb{E}$-product is a Mercer kernel (Theorem 1) with universal approximation (Theorem 4), self-regulation (Proposition 1), and stable gradients (Proposition 2).

Using this kernel in primal form, we construct Neural-Matter Networks (NMNs)—networks where neurons interact through potential fields without requiring Gram matrix inversion. Our contributions: Our contributions span theory, architecture, and interpretability: the $\mathbb{E}$-product eliminates activation functions while maintaining Mercer kernel properties; NMNs reduce memory by 15-25% with infinite differentiability for physics-informed applications; and the geometric structure preserves spatial relationships (Theorems 2, 3), enabling principled analysis of learned representations.

## 2. Methodology: A Framework for Geometry-Aware Computation

The $\mathbb{E}$-product is formally defined as $\mathbb{E}(\mathbf{w}, \mathbf{x}) = \frac{(\mathbf{w}^\top \mathbf{x})^2}{\|\mathbf{w}-\mathbf{x}\|^2 + \varepsilon}$. It exhibits a unique form of non-linearity. Unlike conventional activation functions (e.g., ReLU (Nair & Hinton, 2010), sigmoid) which are often applied as separate, somewhat heuristic, transformations to introduce non-linearity after a linear operation, the non-linearity in the $\mathbb{E}$-product arises directly from its mathematical structure. It is a function of the squared dot product (capturing alignment) and the inverse squared Euclidean distance (capturing proximity) between the weight vector $\mathbf{w}$ and the input vector $\mathbf{x}$. This formulation provides a rich, explainable non-linearity

[1] Anonymous Institution, Anonymous City, Anonymous Region, Anonymous Country. Correspondence to: Anonymous Author <anon.email@domain.com>.

Preliminary work. Under review by the International Conference on Machine Learning (ICML). Do not distribute.

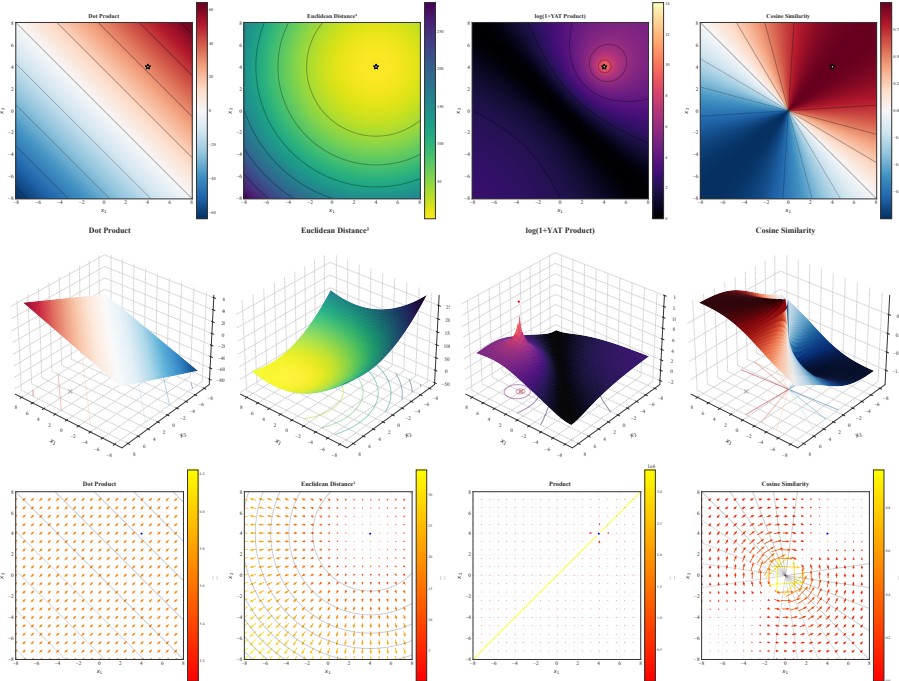

*Figure 1.* Comparison of the gradient field and vector field for Dot Product, Euclidean Distance, Ɛ-product, and Cosine Similarity (from left to right). The heatmaps illustrate how the Ɛ-product, unlike traditional similarity measures, creates a potential well around the weight vector **w**, reflecting both alignment and proximity.

based on fundamental geometric and algebraic relationships, rather than an imposed, "artificial" nonlinear mapping. The interaction between the numerator and the denominator allows for complex responses that are inherently tied to the geometric interplay of the input vectors.

The Ɛ-product creates a potential well around the weight vector **w**, reflecting both alignment and proximity.

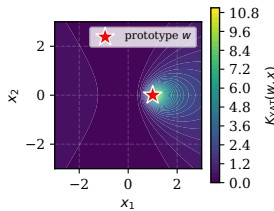

*Figure 2.* Potential well induced by a single Ɛ-neuron. High response occurs near **w** when inputs are both aligned and close, in contrast to unbounded linear hyperplanes.

At initialization, this geometry also exhibits a favorable high-dimensional scaling behavior. Under standard assumptions of i.i.d. zero-mean, constant-variance coordinates for $\mathbf{x}, \mathbf{w} \in \mathbb{R}^d$, both the numerator $A(\mathbf{x}, \mathbf{w}) = (\mathbf{w}^\top \mathbf{x})^2$ and the denominator $r(\mathbf{x}, \mathbf{w}) = \|\mathbf{w} - \mathbf{x}\|^2$ grow linearly with dimension, while their ratio $K(\mathbf{x}, \mathbf{w}) = A/(r + \varepsilon)$ remains $\mathcal{O}(1)$ in expecta-

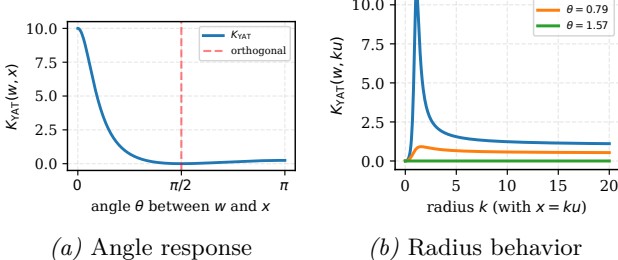

*(a)* Angle response     *(b)* Radius behavior

*Figure 3.* (a) Ɛ-product response as a function of angle $\theta$ between **w** and **x**, demonstrating orthogonality sensitivity ($K_{\text{Ɛ}} = 0$ at $\theta = \pi/2$). (b) Self-regulation property: response converges to $\|\mathbf{w}\|^2 \cos^2 \theta$ as radius $k \to \infty$, ensuring bounded outputs.

tion (Corollary 2). This self-normalizing $\mathcal{O}(1)$ scaling directly counters high-dimensional "saturation" concerns that arise for RBF kernels, whose values vanish exponentially with dimension.

As a Mercer kernel (Theorem 1), on every compact domain $K$ the Ɛ-product admits a unique RKHS (up to isometry) (Theorem 6) and inherits kernel method advantages. Importantly, this kernel is used in its primal form for weight prototype learning and optimization. Consequently, we do not use any Gram matrix, thereby bypassing the stability issues associated with its inversion in dual-form kernel regression (Schölkopf

& Smola, 2002).

When the $\mathbb{E}$-product is applied to probability distributions in the simplex, its extremal values admit an information-geometric characterization:

**Theorem 1** (Mercer property of the $\mathbb{E}$-product kernel)**.** *Let $\varepsilon > 0$ and define*

$$k_{\mathbb{E}}(x, w) = \frac{(x^\top w)^2}{\|x - w\|^2 + \varepsilon}, \qquad x, w \in \mathbb{R}^d.$$

*Then for every compact set $K \subset \mathbb{R}^d$, the kernel $k_{\mathbb{E}}$ is symmetric, continuous, and positive definite on $K$. Consequently, $k_{\mathbb{E}}$ is a Mercer kernel on $K$.*

**Theorem 2** (Minimal Similarity and Statistical Orthogonality)**.** *Let $\mathbf{p}, \mathbf{q} \in \Delta^{n-1}$ be distinct distributions. Then $\mathbb{E}(\mathbf{p}, \mathbf{q}) = 0$ if and only if their supports are disjoint, $\mathrm{supp}(\mathbf{p}) \cap \mathrm{supp}(\mathbf{q}) = \emptyset$. In this case $D_{\mathrm{KL}}(\mathbf{p}\|\mathbf{q}) = \infty$ and the cross-entropy $H(\mathbf{p}, \mathbf{q})$ is infinite.*

**Theorem 3** (Maximal (Singular) Similarity)**.** *Define the $\varepsilon$-dependent $\mathbb{E}$-product*

$$\mathbb{E}_\varepsilon(\mathbf{p}, \mathbf{q}) := \frac{(\mathbf{p}^\top \mathbf{q})^2}{\|\mathbf{p} - \mathbf{q}\|_2^2 + \varepsilon}.$$

*Let $\varepsilon > 0$ and $\mathbf{p}, \mathbf{q} \in \Delta^{n-1}$. Then $\mathbb{E}_\varepsilon(\mathbf{p}, \mathbf{q})$ is finite for all $\mathbf{p}, \mathbf{q}$, and*

$$\mathbb{E}_\varepsilon(\mathbf{p}, \mathbf{p}) = \frac{\|\mathbf{p}\|_2^4}{\varepsilon}.$$

*In the singular limit $\varepsilon \to 0^+$, the self-similarity $\mathbb{E}_\varepsilon(\mathbf{p}, \mathbf{p})$ diverges.*
***Singular joint limit.*** *If $(\mathbf{q}_k)_{k\geq 1} \subset \Delta^{n-1}$ satisfies $\mathbf{q}_k \neq \mathbf{p}$ and $\|\mathbf{q}_k - \mathbf{p}\|_2 \to 0$, and if $\varepsilon_k \to 0^+$, then*

$$\mathbb{E}_{\varepsilon_k}(\mathbf{p}, \mathbf{q}_k) \to \infty.$$

**Corollary 1** (Distributional Identity and KL)**.** *For distributions $\mathbf{p}, \mathbf{q} \in \Delta^{n-1}$, $D_{\mathrm{KL}}(\mathbf{p}\|\mathbf{q}) = 0$ if and only if $\mathbf{p} = \mathbf{q}$ (Gibbs' inequality (Cover & Thomas, 2006)). In this case the cross-entropy reduces to entropy: $H(\mathbf{p}, \mathbf{q}) = H(\mathbf{p})$.*

The $\mathbb{E}$-product creates a potential well around $\mathbf{w}$, where interaction strength diminishes with distance while preserving orientation sensitivity. The explicit gradient structure (Theorem 5) and stable gradient decay (Proposition 2) ensure that gradients vanish for distant inputs, providing natural localization. Input perturbation robustness (Proposition 4) guarantees bounded response changes on bounded domains (with constant controlled by $\varepsilon$). When applied to probability distributions, it connects geometry to information-theoretic extremes (Theorem 2, Theorem 3, Corollary 1).

## 2.1. Neural Matter Network (NMN) Layers

The $\mathbb{E}$-product serves as the foundation for Neural Matter Network layers, employing the non-linear, spatially-aware $\mathcal{K}_{\mathbb{E}}$-kernel as the primary interaction mechanism, replacing conventional linear projections ($\langle \mathbf{w}, \mathbf{x} \rangle$). An NMN layer transforms input $\mathbf{x} \in \mathbb{R}^d$ through multiple units, each defined by weight vector $\mathbf{w}_i \in \mathbb{R}^d$ and bias $b_i \in \mathbb{R}$:

$$h(\mathbf{x}) = \left( s \cdot \sum_{i=1}^n \mathcal{K}_{\mathbb{E}}(\mathbf{w}_i, \mathbf{x}, b_i) \right) = \left( s \cdot \sum_{i=1}^n \frac{(\mathbf{w}_i^\top \mathbf{x} + b_i)^2}{\|\mathbf{w}_i - \mathbf{x}\|^2 + \varepsilon} \right)$$

where $s$ is a scaling factor and $n$ denotes the number of units. Each unit responds based on both alignment and proximity to its learned weight vector, enabling universal function approximation (Theorem 4) as an intrinsic property of the $\mathcal{K}_{\mathbb{E}}$-kernel itself. The self-regulation property (Proposition 1) ensures that outputs remain bounded by encoding an intrinsic normalization in the denominator, rather than relying on separate normalization layers. Figure 4 illustrates the architectural simplification.

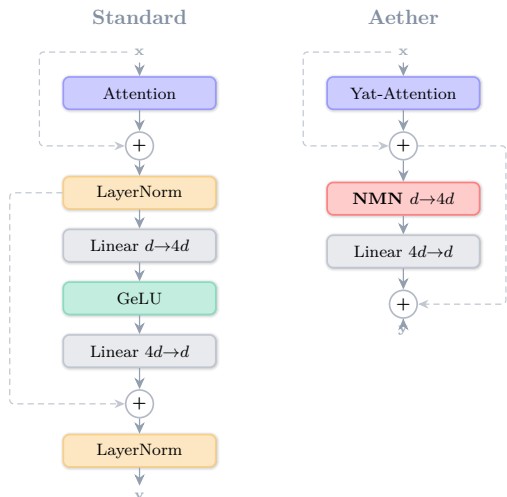

*Figure 4.* Comparison of standard Transformer block (left) and Aether block (right). In Aether-GPT2, $\mathbb{E}$-multi-head attention replaces scaled dot-product attention, and an NMN layer replaces Linear+GeLU, eliminating activation functions and all LayerNorm operations.

**Theorem 4** (Universal approximation with $\mathbb{E}$-kernel)**.** *Let $\mathcal{X} \subset \mathbb{R}^d$ be a compact set. Define the class of functions $\mathcal{F}$ realizable by the network as the linear span of the activation units:*

$$\mathcal{F} = span \left\{ \frac{(\mathbf{x} \cdot \mathbf{w} + b)^2}{\|\mathbf{x} - \mathbf{w}\|^2 + \varepsilon} \;\middle|\; \mathbf{w} \in \mathbb{R}^d, b \in \mathbb{R} \right\}$$

*where $\varepsilon > 0$ is a fixed constant and $b$ is the inner bias parameter. The set $\mathcal{F}$ is dense in $C(\mathcal{X})$ under the uniform norm.*

**Proposition 1** (Natural Self-Regulation). *For any fixed $\mathbf{w}$ and unit direction $\mathbf{u}$, the $\boxed{\text{E}}$-product output remains bounded as $k \to \infty$: $\lim_{k \to \infty} \boxed{\text{E}}(\mathbf{w}, k\mathbf{u}) = \|\mathbf{w}\|^2 \cos^2 \theta$, where $\theta$ is the angle between $\mathbf{w}$ and $\mathbf{u}$. This bounded behavior is visualized in Figure 3(b).*

**Proposition 2** (Gradient Decay for Outliers). *The gradient of the $\boxed{\text{E}}$-product vanishes for distant inputs: $\lim_{\|\mathbf{x}\| \to \infty} \|\nabla_{\mathbf{x}} \boxed{\text{E}}(\mathbf{w}, \mathbf{x})\| = 0$.*

**Theorem 5** (Gradient Direction). *For two vectors $\mathbf{e}_i, \mathbf{e}_j$, the gradient is: $\nabla_{\mathbf{e}_i} \boxed{\text{E}} = \frac{2\langle \mathbf{e}_i, \mathbf{e}_j \rangle}{\varepsilon + \|\mathbf{e}_i - \mathbf{e}_j\|^2} \left( \mathbf{e}_j - \frac{\langle \mathbf{e}_i, \mathbf{e}_j \rangle (\mathbf{e}_i - \mathbf{e}_j)}{\varepsilon + \|\mathbf{e}_i - \mathbf{e}_j\|^2} \right)$.*

**Theorem 6** (RKHS Existence). *For every compact set $K \subset \mathbb{R}^d$, the kernel $k_{\boxed{\text{E}}}$ is positive definite on $K$ (Theorem 1). Hence, by the Moore–Aronszajn theorem, there exists a unique RKHS $\mathcal{H}_K$ and feature map $\phi_K : K \to \mathcal{H}_K$ such that $k_{\boxed{\text{E}}}(\mathbf{x}, \mathbf{y}) = \langle \phi_K(\mathbf{x}), \phi_K(\mathbf{y}) \rangle_{\mathcal{H}_K}$ for all $\mathbf{x}, \mathbf{y} \in K$.*

**Remark 1** (Generalization and Implicit Class Separation). *The RKHS embedding enables explicit generalization bounds: for kernel ridge regression with $n$ samples, the expected error scales as $O(\|f^*\|^2_{\mathcal{H}_{\boxed{\text{E}}}}/\sqrt{n})$ (Theorem 16). Furthermore, the Neural Tangent Kernel of $\boxed{\text{E}}$-networks inherits orthogonality sensitivity: $K_{\boxed{\text{E}}}^{\text{NTK}}(\mathbf{x}, \mathbf{x}') \to 0$ as $\mathbf{x} \perp \mathbf{x}'$ (Theorem 17), providing implicit class separation without explicit contrastive objectives. See Appendix C.19 and C.20 for detailed analysis.*

## 2.2. $\boxed{\text{E}}$-Multi-Head Attention

In Aether-GPT2 we also replace scaled dot-product attention (Vaswani et al., 2017) with a $\boxed{\text{E}}$-based multi-head attention mechanism. For a sequence of length $L$ with per-head query, key, and value matrices $Q, K, V \in \mathbb{R}^{L \times d_h}$, we define the $\boxed{\text{E}}$-similarity between a query $q_i$ and key $k_j$ as

$$K_{\boxed{\text{E}}}(q_i, k_j) = \frac{(q_i^\top k_j)^2}{\|q_i - k_j\|_2^2 + \varepsilon}.$$

This induces an attention score matrix $S \in \mathbb{R}^{L \times L}$ with entries

$$S_{ij} = K_{\boxed{\text{E}}}(q_i, k_j),$$

and the single-head $\boxed{\text{E}}$-attention is given by

$$\text{YATAttn}(Q, K, V) = \text{softmax}_j(S_{ij}) V,$$

where the softmax is taken row-wise over keys $j$ for each query position $i$. In the multi-head setting, we use learned projections $W_Q^{(h)}, W_K^{(h)}, W_V^{(h)}$ per head $h$,

$$Q^{(h)} = X W_Q^{(h)}, \quad K^{(h)} = X W_K^{(h)}, \quad V^{(h)} = X W_V^{(h)},$$

apply the above $\boxed{\text{E}}$-attention independently to each head, and concatenate the outputs:

$$\boxed{\text{E}}\text{-MHA}(X) = \text{Concat}_h \big( \text{YATAttn}(Q^{(h)}, K^{(h)}, V^{(h)}) \big) W^O.$$

This preserves the expressive power of multi-head attention while endowing the similarity measure with the same potential-well geometry and self-regularization properties as the NMN layers.

**Proposition 3** (Lipschitz Continuity). *Fix $\varepsilon > 0$ and a weight vector $\mathbf{w}$ with $\|\mathbf{w}\|_2 \leq 1$. Then the map $\mathbf{x} \mapsto \boxed{\text{E}}(\mathbf{w}, \mathbf{x})$ is Lipschitz continuous on the unit ball $\{\mathbf{x} \in \mathbb{R}^d : \|\mathbf{x}\|_2 \leq 1\}$ with Lipschitz constant $L = 2/\varepsilon + 4/\varepsilon^2$.*

**Lemma 1** (Analyticity). *For $\varepsilon > 0$, the map $\mathbf{x} \mapsto \boxed{\text{E}}(\mathbf{w}, \mathbf{x})$ is real-analytic on $\mathbb{R}^d$ (infinitely differentiable).*

**Proposition 4** (Input Perturbation Robustness). *Fix $\varepsilon > 0$ and $\|\mathbf{w}\|_2 \leq 1$. For any $\mathbf{x}, \mathbf{x}'$ in the unit ball with $\|\mathbf{x}' - \mathbf{x}\|_2 \leq \delta$,*

$$|\boxed{\text{E}}(\mathbf{w}, \mathbf{x}') - \boxed{\text{E}}(\mathbf{w}, \mathbf{x})| \leq \left( \frac{2}{\varepsilon} + \frac{4}{\varepsilon^2} \right) \delta.$$

**Corollary 2** (Dimensional Self-Normalization). *Let $\mathbf{x}, \mathbf{w} \in \mathbb{R}^d$ have i.i.d. zero-mean coordinates with $\text{Var}(x_i) = \text{Var}(w_i) = \sigma^2$ (constant in $d$), and assume in addition that:*

- *$\mathbf{x}$ and $\mathbf{w}$ are independent, and*

- *the coordinates are sub-Gaussian with parameter independent of $d$ (e.g., Gaussian initialization), hence have finite fourth moments.*

*Fix $\varepsilon > 0$. Then as $d \to \infty$,*

$$\mathbb{E}[\boxed{\text{E}}(\mathbf{w}, \mathbf{x})] = \mathcal{O}(1).$$

**Proposition 5** (Extremal Similarity on the Simplex). *For $\mathbf{p}, \mathbf{q} \in \Delta^{n-1}$: $\boxed{\text{E}}(\mathbf{p}, \mathbf{q}) = 0$ iff $\text{supp}(\mathbf{p}) \cap \text{supp}(\mathbf{q}) = \emptyset$, which implies $\text{KL}(\mathbf{p} \| \mathbf{q}) = \infty$.*

**Remark 2** (Optimal $\varepsilon$ Scaling). *For noisy inputs with $\mathbf{n} \sim \mathcal{N}(0, \sigma^2 \mathbf{I})$, the stability constant should scale as $\varepsilon^* \propto d\sigma^2$ to maximize gradient signal-to-noise ratio.*

## 2.3. Architectural Implementation

Following the representer theorem (Schölkopf et al., 2001), optimal solutions in kernel methods lie in the span of kernel evaluations at training points. Since the $\boxed{\text{E}}$-product is a Mercer kernel, composing multiple $\boxed{\text{E}}$-layers without intervening linear projections

would create a deep kernel that loses this representational guarantee. Our architecture therefore pairs each $\mathtt{E}$-kernel layer with a subsequent linear projection, preserving the kernel's theoretical properties while enabling depth. We also eliminate normalization layers entirely: the $\mathtt{E}$-product's self-regulation property (Proposition 1) provides intrinsic normalization, making explicit batch or layer normalization redundant and potentially harmful to gradient flow. The Lipschitz regularity (Proposition 3) and analyticity (Lemma 1) ensure stable training dynamics and infinite differentiability. All NMN-based layers use the adaptive scaling factor $s = \left( \frac{n}{\log(1+n)} \right)^{\alpha}$, where $n$ is the number of units and $\alpha$ is a learnable parameter initialized at 1.

**Computational Efficiency:** The $\mathtt{E}$-product layer maintains $\Theta(Bnd)$ computational complexity identical to standard linear layers while providing 15-25% memory reduction through elimination of activation storage. Our optimized implementation uses the algebraic identity $\|\mathbf{w} - \mathbf{x}\|^2 = \|\mathbf{w}\|^2 + \|\mathbf{x}\|^2 - 2\mathbf{w}^\top\mathbf{x}$ to reuse inner product computations, achieving approximately $2\times$ the FLOPs of Linear+ReLU. The approach offers natural numerical stability and becomes increasingly efficient at larger layer sizes, making it particularly suitable for large-scale applications.

# 3. Results and Discussion

We evaluate the $\mathtt{E}$-product on three tasks: XOR separability (demonstrating non-linearity), MNIST classification (prototype learning), and language modeling (Aether-GPT2).

## 3.1. XOR Separability with a Single Unit

The $\mathtt{E}$-product's inherent non-linearity enables solving non-linearly separable problems with a single unit. For XOR with inputs $(0,0) \to 0$, $(0,1) \to 1$, $(1,0) \to 1$, $(1,1) \to 0$, a single $\mathtt{E}$-product unit with $\mathbf{w} = [1, -1]^\top$ achieves perfect separation:

| $\mathbf{x}$ | $\mathbf{w}^\top\mathbf{x}$ | $\mathcal{K}_{\mathtt{E}}(\mathbf{w}, \mathbf{x})$ | Class |
|---|---|---|---|
| $(0,0)$ | $0$ | $0$ | $0$ |
| $(0,1)$ | $-1$ | $1/(5 + \varepsilon) > 0$ | $1$ |
| $(1,0)$ | $1$ | $1/(1 + \varepsilon) > 0$ | $1$ |
| $(1,1)$ | $0$ | $0$ | $0$ |

By definition (Eq. (1)), $\mathbf{w}^\top\mathbf{x} = 0$ implies $\mathcal{K}_{\mathtt{E}}(\mathbf{w}, \mathbf{x}) = 0$ (in particular when $\mathbf{w} \perp \mathbf{x}$). See Appendix D for formal proof.

## 3.2. Decision Boundaries and Localization

Unlike linear neurons that induce unbounded hyperplane partitions, $\mathtt{E}$-product neurons generate localized decision surfaces around prototypes. This follows from self-regulation (Proposition 1): the response $\mathcal{K}_{\mathtt{E}}(\mathbf{w}, \mathbf{x}) \to \|\mathbf{w}\|^2 \cos^2 \theta$ as $\|\mathbf{x}\| \to \infty$.

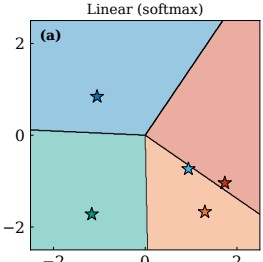 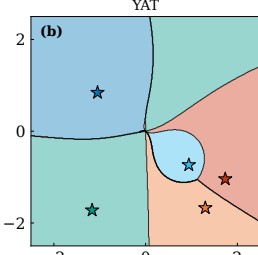

*Figure 5.* Decision boundaries in 2D: linear (left) creates unbounded half-spaces; $\mathtt{E}$-product (right) forms localized regions around prototypes (stars).

The extremal similarity results characterize boundary cases. Orthogonality ($\mathbf{w}^\top\mathbf{x} = 0$) yields $\mathcal{K}_{\mathtt{E}} = 0$ directly from Eq. (1). Identity ($\mathbf{w} = \mathbf{x}$) yields $\mathcal{K}_{\mathtt{E}}(\mathbf{w}, \mathbf{w}) = \|\mathbf{w}\|_2^4/\varepsilon$ (Theorem 3 states this on the simplex; the same algebra holds in $\mathbb{R}^d$). Lipschitz continuity (Proposition 3) ensures smooth interpolation.

## 3.3. MNIST Classification

We compare a 10-neuron $\mathtt{E}$-product classifier against a linear baseline on MNIST (60k training, 10k test samples). Architecture: $C = 10$ prototypes $\mathbf{w}_i \in \mathbb{R}^{784}$. Training: Adam (lr=0.001), 5 epochs. Baseline: linear classifier $z_i = \mathbf{w}_i^\top\mathbf{x}$ with softmax.

*Table 1.* MNIST results (10-neuron classifier).

|  | Acc. | $\Delta\|\mathbf{w}\|$ | $\alpha$ |
|---|---|---|---|
| Linear | 92.08% | +13.8% | – |
| $\mathtt{E}$ | **92.38%** | $-4.5\%$ | $1 \to 2.68$ |

**Bounded Prototype Evolution.** The self-regulation property (Proposition 1) predicts stable prototype magnitudes. Empirically, linear prototypes grow unboundedly (+13.8%), while $\mathtt{E}$-product prototypes contract slightly ($-4.5\%$), confirming bounded response fields. The learnable scaling factor $\alpha$ (initialized at 1) increases to 2.68, amplifying bounded $\mathtt{E}$-responses for softmax discrimination.

**Superposition Robustness.** The squared numerator $(\mathbf{w}^\top\mathbf{x})^2$ creates approximate invariance under sign flip. Prototype inversion ($\mathbf{w} \to -\mathbf{w}$) yields:

**Learned Prototypes: Linear vs YAT**

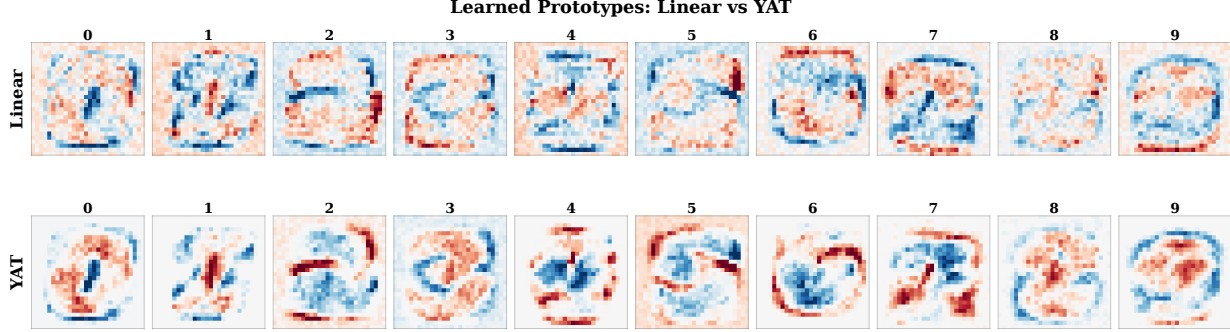

*Figure 6.* Learned MNIST prototypes from a linear classifier (top) and an E-product classifier (bottom). We visualize the learned class prototypes as 28×28 weight images using a shared diverging colormap and robust limits for fair comparison across classes and methods. E-product prototypes tend to exhibit more localized, structured strokes, consistent with the kernel's potential-well geometry.

|  | Original | Inverted |
|---|---|---|
| Linear | 92.04% | 0.01% |
| E | 92.18% | 87.87% |

For linear neurons, $(-\mathbf{w})^\top \mathbf{x} = -\mathbf{w}^\top \mathbf{x}$ flips the logit sign, causing catastrophic failure. For E-product, the numerator invariance provides robustness.

**Territorial Structure.** Since the numerator is $(\mathbf{w}_i^\top \mathbf{w}_j)^2$, orthogonal prototypes satisfy $\mathbf{E}(\mathbf{w}_i, \mathbf{w}_j) = 0$. The E-product develops heterogeneous structure: high similarity for morphologically similar digits (7-9), sharp boundaries elsewhere.

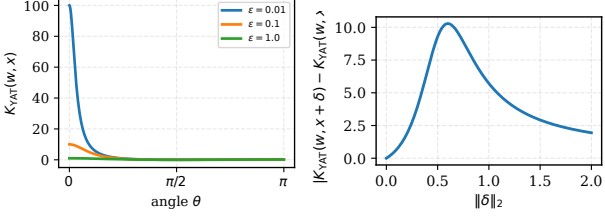

*Figure 7.* (a) Effect of regularization parameter $\varepsilon$ on E-product shape: smaller $\varepsilon$ sharpens the response peak. (b) Input perturbation robustness: bounded change in output for bounded input perturbations, demonstrating Lipschitz continuity.

### 3.4. Extreme Classification Benchmark

We evaluate the E-product on the Extreme Classification Benchmark using the Eurlex-4K dataset. As shown in Table 2, the E-product classifier outperforms the standard inner product classifier on the main accuracy metrics: P@1 (0.6465 vs 0.6235), P@3 (0.5114 vs 0.5041), and P@5 (0.4271 vs 0.4125). While the inner product achieves higher values on the propensity-scored metrics PSP@1–5, the E-product attains stronger overall predictive accuracy on extreme

classification, highlighting its robustness for top-label retrieval tasks.

### 3.5. Language Modeling: Aether-GPT2

We train Aether-GPT2 (124M parameters) on 2.5B tokens from FineWeb. The architectural modifications follow Section 2.1 and 2.2: the MLP block is replaced by an NMN layer (dim 3072, i.e., 4× hidden dim) followed by a linear projection, and scaled dot-product attention is replaced by E-attention. Rather than being "normalization-free", Aether-GPT2 relies on the intrinsic normalization encoded in the E-product denominator and its self-regulation property, so we do not add explicit LayerNorm in these blocks.

**Architectural Simplification.** The self-regulation property (Proposition 1) and dimensional scaling (Corollary 2) eliminate the need for layer normalization. This yields 15–25% memory reduction by removing activation storage.

**Mixed-Precision Stability.** All language-modeling runs reported here were executed in BF16. The bounded E-product response provides numerical stability without explicit normalization (see Appendix E.6 for run provenance and configuration). The RKHS embedding (Theorem 6) ensures well-defined feature spaces.

See Appendix E.6 for detailed configuration and BF16 results.

### 3.6. Ablation: Layer Normalization Incompatibility

A critical design choice in Aether architectures is the *removal* of layer normalization. We find that reintroducing LayerNorm to Aether-GPT2 causes training

|  | P@k | | | PSP@k | | |
| --- | --- | --- | --- | --- | --- | --- |
| Method | P@1 | P@3 | P@5 | PSP@1 | PSP@3 | PSP@5 |
| Ɛ-product | **0.6465** | **0.5114** | **0.4271** | **1.1542** | 0.9664 | 0.8443 |
| Inner Product | 0.6235 | 0.5041 | 0.4125 | 1.2117 | **1.0215** | **0.8587** |

*Table 2.* Extreme classification results on Eurlex-4K. Ɛ-product improves top-$k$ precision (P@1–5), while the inner product baseline achieves slightly higher propensity-scored precision (PSP@3, PSP@5).

*Table 3.* GPT-2 vs Aether-GPT2 (2.5B tokens, identical hyperparameters).

|  | GPT-2 | Aether |
| --- | --- | --- |
| Activation | GeLU | Ɛ |
| LayerNorm | Yes | **No** |
| Train Loss (Final) | 4.1969 | **4.0479** |
| Val Loss (Final) | 4.6417 | **4.5747** |
| Val improvement | – | **1.45%** |
| Throughput (tokens/s) | comparable | comparable or higher |
| Memory | – | $-15$–$25\%$ |

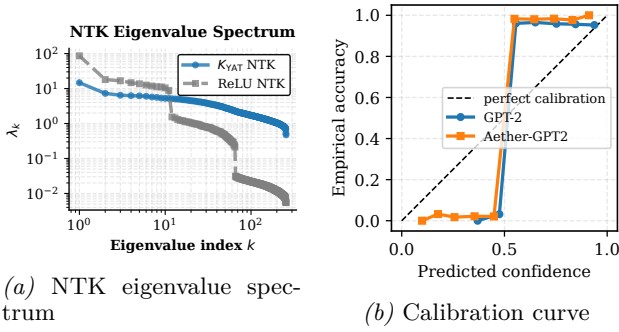

*(a)* NTK eigenvalue spectrum

*(b)* Calibration curve

*Figure 8.* (a) Eigenvalue spectrum of Ɛ-NTK vs ReLU-NTK: faster decay ($O(k^{-2/d})$) indicates smoother function spaces. (b) Reliability diagram comparing GPT-2 and Aether-GPT2: Aether exhibits better calibration, with predictions closer to the diagonal.

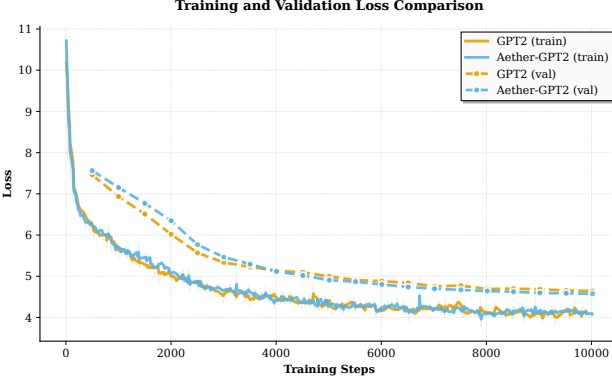

*Figure 9.* Training and validation loss curves (W&B). Solid: training; dashed: validation.

collapse, with validation loss diverging within the first 1000 steps.

**Implicit Normalization in the Ɛ-Product.** The denominator of the Ɛ-product already encodes norm information through the squared distance expansion:

$$\|\mathbf{w} - \mathbf{x}\|^2 = \|\mathbf{w}\|^2 + \|\mathbf{x}\|^2 - 2\mathbf{w}^\top \mathbf{x}. \qquad (2)$$

Thus the full Ɛ-product can be rewritten as:

$$\mathbb{E}(\mathbf{w}, \mathbf{x}) = \frac{(\mathbf{w}^\top \mathbf{x})^2}{\|\mathbf{w}\|^2 + \|\mathbf{x}\|^2 - 2\mathbf{w}^\top \mathbf{x} + \varepsilon}.$$

The denominator explicitly depends on $\|\mathbf{w}\|^2$ and $\|\mathbf{x}\|^2$, providing intrinsic scale awareness. When inputs are pre-normalized via LayerNorm (forcing $\|\mathbf{x}\|^2 \approx d$ and removing variance information), the denominator loses its adaptive regularization: all inputs become equidistant from the origin, collapsing the proximity structure that the Ɛ-product exploits.

**Redundancy and Interference.** Layer normalization serves two purposes in standard architectures: (1) variance stabilization for gradient flow, and (2) preventing internal covariate shift (Ioffe & Szegedy, 2015). The self-regulation property (Proposition 1) already provides bounded outputs independent of input scale. Moreover, the dimensional scaling corollary (Corollary 2) ensures $\mathcal{O}(1)$ expected values at initialization. LayerNorm therefore introduces redundant normalization that interferes with the geometric structure the Ɛ-product relies upon.

*Table 4.* Ablation: LayerNorm impact on Aether-GPT2.

| Configuration | Val Loss | Status |
| --- | --- | --- |
| Aether (no LayerNorm) | **4.5747** | Converged |
| Aether + Pre-LN | >10 | Diverged |
| Aether + Post-LN | >10 | Diverged |

This ablation confirms that the architectural simplification in Aether is not merely optional—layer normalization is *incompatible* with the Ɛ-product's geometric foundations.

## 4. Related Work

### 4.1. Kernel Methods and Neural Tangent Kernels

Kernel methods enable non-linear learning through implicit feature mappings (Schölkopf & Smola, 2002). SVMs (Cortes, 1995) and kernel PCA (Schölkopf et al., 1998) established the foundation, with Gaussian Processes (Williams & Rasmussen, 2006) extending to probabilistic inference. Scalability came through the Nyström method (Williams & Seeger, 2000) and Random Fourier Features (Rahimi & Recht, 2007).

The Neural Tangent Kernel (Jacot et al., 2018) bridges kernels and deep learning by characterizing infinite-width networks as linear models under gradient descent (Lee et al., 2019; Arora et al., 2019). Since the E-product is a valid Mercer kernel (Theorem 1), NTK theory extends to our framework (Proposition 6), enabling infinite-width analysis of geometric operators. The connection between SGD and kernel learning (Daniely, 2017; Li & Liang, 2018) further supports our approach.

Distance-based kernels (RBF) emphasize proximity; polynomial kernels capture feature interactions. The E-product unifies both: the squared numerator provides polynomial-like alignment, while the inverse-square denominator gives RBF-like locality with intrinsic self-regularization.

Deep kernel learning (Wilson et al., 2016; Aitchison et al., 2021) combines neural networks with kernel flexibility, but operates in dual form requiring $O(n^2)$ Gram matrices. Our primal-form approach computes directly in feature space, avoiding this cost. Prior kernelized networks (Cho & Saul, 2009; Mairal et al., 2014) approximate kernels within linear-then-activate structures; the E-product is simultaneously the computational primitive and the kernel.

### 4.2. Alternative Neural Operators

Quadratic neurons (Fan et al., 2020; Liao et al., 2024) achieve non-linearity through polynomial forms but ignore geometric structure. Multiplicative interactions (Jayakumar et al., 2020) and gated linear units (Dauphin et al., 2016) introduce element-wise products yet retain activation dependence. SIREN (Sitzmann et al., 2020) and Fourier feature networks (Tancik et al., 2020) employ periodic activations for implicit representations.

The E-product differs fundamentally: it integrates alignment (squared dot product) and proximity (inverse distance) into a single operator, achieving non-linearity through geometric structure rather than functional composition—no activation functions required.

### 4.3. Geometric Foundations

The inverse-square law governs fundamental interactions across physics: gravitation (Newton, 1687), electrostatics (de Coulomb, 1785), and radiation (Gauss, 1835). This principle—intensity scaling inversely with squared distance—appears in engineering (signal propagation (Rappaport, 2002)) and information theory (Tanimoto similarity (Tanimoto, 1958)). Geometric deep learning (Bronstein et al., 2021) provides a unifying framework for exploiting such structure in neural architectures.

The E-product operationalizes this geometric principle for neural computation: interaction strength grows with alignment but decays with distance, providing a physics-inspired foundation for learning representations.

## 5. Conclusion

We introduced the E-product, a kernel operator that unifies alignment and proximity: $\mathcal{K}_E(\mathbf{w}, \mathbf{x}) = (\mathbf{w}^\top \mathbf{x})^2/(\|\mathbf{w} - \mathbf{x}\|^2 + \varepsilon)$. We proved it is a valid Mercer kernel with analyticity, Lipschitz continuity on bounded domains, self-regulation, and gradient decay—properties that enable Neural Matter Networks to preserve universal approximation while using intrinsic normalization via the denominator instead of separate normalization layers. Empirically, Aether-GPT2 achieves lower validation loss than GPT-2 with a comparable parameter budget while replacing both scaled dot-product attention and MLP blocks by E-based mechanisms. By grounding neural computation in physics-inspired geometry, this work offers a principled path toward simpler, more interpretable architectures.

## License

This work is licensed under the Affero GNU General Public License (AGPL) v3.0. The AGPL is a free software license that ensures end users have the freedom to run, study, share, and modify the software. It requires that any modified versions of the software also be distributed under the same license, ensuring that the freedoms granted by the original license are preserved in derivative works. The full text of the AGPL v3.0 can be found at https://www.gnu.org/licenses/agpl-3.0.en.html. By using this work, you agree to comply with the terms of the AGPL v3.0.

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

## A. Appendix

## B. Squashing Functions for Non-Negative Scores

The non-negative nature of $\mathbb{E}$-product scores necessitates specialized normalization functions. We categorize these into *competitive* (vector-normalizing) and *individualistic* (element-wise) functions.

### B.1. Competitive Normalization

Competitive functions induce coupling between dimensions, interpreting scores as relative strengths within a distribution.

**Softermax.** A generalized normalization function for non-negative scores $\mathbf{x} \in \mathbb{R}_{\geq 0}^d$:

$$\text{softermax}_n(x_k, \{\mathbf{x}\}) = \frac{x_k^n}{\epsilon + \sum_{i=1}^d x_i^n}, \tag{3}$$

where $n > 0$ controls the distribution sharpness (analogous to inverse temperature) and $\epsilon > 0$ ensures numerical stability and prevents division by zero for sparse inputs. Unlike softmax, this formulation avoids exponential terms, improving numerical stability for large input magnitudes.

### B.2. Individualistic Squashing

Individualistic functions map scores to bounded intervals element-wise, preserving independence.

**Soft-Sigmoid.** Maps $x \in [0, \infty)$ to $[0, 1)$:

$$\sigma_n(x) = \frac{x^n}{1 + x^n}. \tag{4}$$

This algebraic sigmoid provides a heavy-tailed alternative to the logistic sigmoid, with polynomial rather than exponential saturation.

**Soft-Tanh.** Maps $x \in [0, \infty)$ to $[-1, 1)$:

$$\tau_n(x) = \frac{x^n - 1}{x^n + 1}. \tag{5}$$

This corresponds to a rescaled soft-sigmoid: $\tau_n(x) = 2\sigma_n(x) - 1$. The parameter $n$ acts as a gain factor, controlling the steepness of the transition from the distinct states $-1$ (orthogonality/dissimilarity) to $+1$ (alignment/similarity).

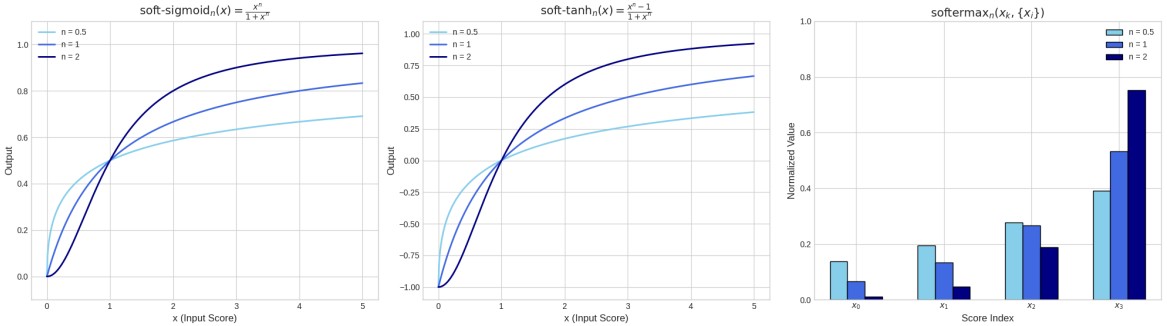

*Figure 10.* Algebraic squashing functions for non-negative $\mathbb{E}$-product scores. These offer bounded, monotonic mappings without exponential saturation.

### B.3. Mathematical Preliminaries

This section collects the key mathematical tools and terminology used throughout the paper.

### B.3.1. KERNEL TERMINOLOGY

**Definition 1** (Kernel). *A kernel is a function $k : \mathcal{X} \times \mathcal{X} \to \mathbb{R}$ that measures similarity between two inputs (Schölkopf & Smola, 2002).*

**Definition 2** (Gram Matrix). *Given points $x_1, \ldots, x_n \in \mathcal{X}$, the Gram matrix of $k$ is $K \in \mathbb{R}^{n \times n}$ with entries $K_{ij} = k(x_i, x_j)$.*

**Definition 3** (Positive Definite Kernel). *A symmetric kernel $k$ is positive definite (PD) if its Gram matrix is positive semidefinite for every finite set of points:*

$$\sum_{i,j=1}^{n} a_i a_j \, k(x_i, x_j) \geq 0 \quad \text{for all } a \in \mathbb{R}^n.$$

**Definition 4** (Feature Map and RKHS). *A feature map is a map $\Phi : \mathcal{X} \to \mathcal{H}$ such that $k(x, y) = \langle \Phi(x), \Phi(y) \rangle_{\mathcal{H}}$. The reproducing kernel Hilbert space (RKHS) of a PD kernel is the Hilbert space where the kernel becomes an inner product (Aronszajn, 1950).*

### B.3.2. CLOSURE PROPERTIES OF PD KERNELS

**Theorem 7** (PD Closure Properties). *If $k_1, k_2$ are positive definite kernels on $\mathcal{X}$, then (Schölkopf & Smola, 2002):*

1. *$k_1 + k_2$ is PD (closure under addition)*

2. *$w \cdot k_1$ is PD for $w \geq 0$ (closure under nonnegative scaling)*

3. *$k_1 \cdot k_2$ is PD (Schur product theorem)*

4. *$\int k_s \, w(s) \, ds$ is PD for $w(s) \geq 0$ (closure under nonnegative mixtures)*

*Context: These closure properties are used in the Mercer kernel proof (Theorem 1) to show that the product of the polynomial kernel and inverse multiquadric is PD.*

### B.3.3. LAPLACE TRANSFORM IDENTITY

**Theorem 8** (Laplace Identity for $1/y$). *For any $y > 0$:*

$$\frac{1}{y} = \int_0^{\infty} e^{-sy} \, ds.$$

*Context: This identity converts the rational term $1/(\|\mathbf{x} - \mathbf{w}\|^2 + \varepsilon)$ into an integral over exponentials, enabling the inverse multiquadric to be expressed as a nonnegative mixture of Gaussian kernels.*

### B.3.4. COMPLETE MONOTONICITY AND BERNSTEIN REPRESENTATION

**Theorem 9** (Bernstein's Theorem). *A function $f : [0, \infty) \to \mathbb{R}$ is completely monotonic (i.e., $(-1)^n f^{(n)}(y) \geq 0$ for all $n \geq 0$) if and only if it is the Laplace transform of a non-negative measure (Schilling et al., 2012):*

$$f(t) = \int_0^{\infty} e^{-ts} \, d\mu(s), \quad \mu \geq 0.$$

*Context: The function $1/y$ is completely monotone, so it admits a nonnegative exponential-mixture representation. This justifies the decomposition used in the Mercer proof.*

### B.3.5. INTEGRAL EXCHANGE (TONELLI-FUBINI)

**Theorem 10** (Tonelli-Fubini). *For $\sigma$-finite measure spaces and measurable $f \geq 0$ (Folland, 1999):*

$$\int_{X \times Y} f \, d(\mu \times \nu) = \int_X \left( \int_Y f \, d\nu \right) d\mu = \int_Y \left( \int_X f \, d\mu \right) d\nu.$$

*Context: Justifies exchanging integrals and sums in PD kernel proofs: "integral of PD kernels is PD" requires moving the integral outside the quadratic form.*

### B.3.6. Information Theory

**Theorem 11** (Gibbs' Inequality)**.** *For probability distributions $P, Q$ (Cover & Thomas, 2006):*

$$D_{KL}(P\|Q) = \sum_x P(x) \log \frac{P(x)}{Q(x)} \geq 0,$$

*with equality iff $P = Q$.*

*Context: Referenced in Corollary 1 connecting distributional identity to vanishing KL divergence.*

### B.3.7. Harmonic Analysis

**Theorem 12** (Bochner's Theorem)**.** *A continuous function $k : \mathbb{R}^d \to \mathbb{C}$ is positive definite and translation-invariant iff it is the Fourier transform of a finite non-negative measure (Rudin, 1991).*

**Theorem 13** (Hahn-Banach Density Criterion)**.** *A linear subspace $M \subset V$ is dense in $V$ iff every continuous linear functional vanishing on $M$ vanishes on $V$ (Rudin, 1991).*

*Context: Used in the universal approximation proof (Theorem 4).*

## C. Proofs of Main Theorems

This section provides the proofs for theorems stated in the main body.

### C.1. Proof of Theorem 5 (Gradient Direction)

Let $s = \langle \mathbf{e}_i, \mathbf{e}_j \rangle$ and $D = \varepsilon + \|\mathbf{e}_i - \mathbf{e}_j\|^2$. Then $\mathbb{E} = s^2/D$.

Using the quotient rule:

$$\nabla_{\mathbf{e}_i} \mathbb{E} = \frac{2s \cdot \nabla_{\mathbf{e}_i} s \cdot D - s^2 \cdot \nabla_{\mathbf{e}_i} D}{D^2}.$$

We have $\nabla_{\mathbf{e}_i} s = \mathbf{e}_j$ and $\nabla_{\mathbf{e}_i} D = 2(\mathbf{e}_i - \mathbf{e}_j)$. Substituting:

$$\nabla_{\mathbf{e}_i} \mathbb{E} = \frac{2s\mathbf{e}_j \cdot D - 2s^2(\mathbf{e}_i - \mathbf{e}_j)}{D^2} = \frac{2s}{D} \left( \mathbf{e}_j - \frac{s(\mathbf{e}_i - \mathbf{e}_j)}{D} \right). \quad \square$$

### C.2. Proof of Proposition 3 (Lipschitz Continuity)

Fix $\varepsilon > 0$ and assume $\|\mathbf{x}\|_2 \leq 1$ and $\|\mathbf{w}\|_2 \leq 1$. Let $s = \langle \mathbf{x}, \mathbf{w} \rangle$ and $D = \varepsilon + \|\mathbf{x} - \mathbf{w}\|_2^2$. Then $D \geq \varepsilon$. From the gradient formula (Theorem 5):

$$\|\nabla_{\mathbf{x}} \mathbb{E}\| \leq \frac{2|s|}{D} \left( \|\mathbf{w}\|_2 + \frac{|s| \cdot \|\mathbf{x} - \mathbf{w}\|_2}{D} \right).$$

For $\|\mathbf{x}\|_2, \|\mathbf{w}\|_2 \leq 1$, we have $|s| = |\langle \mathbf{x}, \mathbf{w} \rangle| \leq 1$ and $\|\mathbf{x} - \mathbf{w}\|_2 \leq 2$, so:

$$\|\nabla_{\mathbf{x}} \mathbb{E}\| \leq \frac{2}{\varepsilon} \left( 1 + \frac{2}{\varepsilon} \right) = \frac{2}{\varepsilon} + \frac{4}{\varepsilon^2}. \quad \square$$

### C.3. Proof of Proposition 5 (Extremal Similarity)

(1) The numerator $\langle \mathbf{p}, \mathbf{q} \rangle^2 = \left( \sum_i p_i q_i \right)^2 = 0$ if and only if all terms $p_i q_i = 0$, which occurs precisely when supports are disjoint.

(2) If $\mathrm{supp}(\mathbf{p}) \cap \mathrm{supp}(\mathbf{q}) = \emptyset$, there exists $i$ with $p_i > 0$ and $q_i = 0$, making $\mathrm{KL}(\mathbf{p}\|\mathbf{q}) = \sum_i p_i \log(p_i/q_i) = \infty$.

**Cross-entropy (used in Theorem 2).** Under the same condition, the cross-entropy

$$H(\mathbf{p}, \mathbf{q}) = -\sum_i p_i \log q_i$$

contains a term with $p_i > 0$ and $q_i = 0$, hence $-\log q_i = -\log 0 = +\infty$ and therefore $H(\mathbf{p}, \mathbf{q}) = \infty$ under the standard convention $\log 0 = -\infty$. □

## C.4. Proof of Theorem 2 (Minimal Similarity and Statistical Orthogonality)

Let $\mathbf{p}, \mathbf{q} \in \Delta^{n-1}$ and $\varepsilon > 0$. Since $\|\mathbf{p} - \mathbf{q}\|_2^2 + \varepsilon > 0$, we have $\mathbb{E}(\mathbf{p}, \mathbf{q}) = 0$ if and only if $\mathbf{p}^\top \mathbf{q} = 0$. On the simplex, $p_i q_i \geq 0$ for all $i$, so $\sum_i p_i q_i = 0$ holds if and only if $p_i q_i = 0$ for all $i$, i.e. $\operatorname{supp}(\mathbf{p}) \cap \operatorname{supp}(\mathbf{q}) = \emptyset$. In this case $D_{\mathrm{KL}}(\mathbf{p}\|\mathbf{q}) = \infty$ by the argument in Proposition 5, and $H(\mathbf{p}, \mathbf{q}) = \infty$ by the cross-entropy note above. □

## C.5. Proof of Theorem 3 (Maximal (Singular) Similarity)

Fix $\varepsilon > 0$ and define $\mathbb{E}_\varepsilon(\mathbf{p}, \mathbf{q}) := \frac{(\mathbf{p}^\top \mathbf{q})^2}{\|\mathbf{p}-\mathbf{q}\|_2^2+\varepsilon}$. For any $\mathbf{p}, \mathbf{q} \in \Delta^{n-1}$, the denominator is at least $\varepsilon$, hence $\mathbb{E}_\varepsilon(\mathbf{p}, \mathbf{q})$ is finite. If $\mathbf{q} = \mathbf{p}$, then $\|\mathbf{p} - \mathbf{q}\|_2^2 = 0$ and

$$\mathbb{E}_\varepsilon(\mathbf{p}, \mathbf{p}) = \frac{(\mathbf{p}^\top \mathbf{p})^2}{\varepsilon} = \frac{\|\mathbf{p}\|_2^4}{\varepsilon},$$

so $\mathbb{E}_\varepsilon(\mathbf{p}, \mathbf{p}) \to \infty$ as $\varepsilon \to 0^+$.

**Singular joint limit.** Let $(\mathbf{q}_k)_{k\geq 1} \subset \Delta^{n-1}$ satisfy $\mathbf{q}_k \neq \mathbf{p}$ and $\|\mathbf{q}_k - \mathbf{p}\|_2 \to 0$, and let $\varepsilon_k \to 0^+$. Then $(\mathbf{p}^\top \mathbf{q}_k)^2 \to (\mathbf{p}^\top \mathbf{p})^2 = \|\mathbf{p}\|_2^4 > 0$ while $\|\mathbf{p} - \mathbf{q}_k\|_2^2 + \varepsilon_k \to 0$. Therefore $\mathbb{E}_{\varepsilon_k}(\mathbf{p}, \mathbf{q}_k) \to \infty$. □

## C.6. Proof of Corollary 1 (Distributional Identity and KL)

By Gibbs' inequality (see e.g. Cover & Thomas (2006)), $D_{\mathrm{KL}}(\mathbf{p}\|\mathbf{q}) \geq 0$ with equality if and only if $\mathbf{p} = \mathbf{q}$. When $\mathbf{p} = \mathbf{q}$, the cross-entropy satisfies $H(\mathbf{p}, \mathbf{q}) = H(\mathbf{p})$. □

## C.7. Proof of Proposition 1 (Self-Regulation)

For $\mathbf{x} = k\mathbf{u}$:

$$\mathbb{E}(\mathbf{w}, k\mathbf{u}) = \frac{(k\mathbf{w}^\top \mathbf{u})^2}{\|\mathbf{w} - k\mathbf{u}\|^2 + \varepsilon} = \frac{k^2(\mathbf{w}^\top \mathbf{u})^2}{\|\mathbf{w}\|^2 - 2k\mathbf{w}^\top \mathbf{u} + k^2 + \varepsilon}.$$

Dividing numerator and denominator by $k^2$:

$$\mathbb{E}(\mathbf{w}, k\mathbf{u}) = \frac{(\mathbf{w}^\top \mathbf{u})^2}{\|\mathbf{w}\|^2/k^2 - 2\mathbf{w}^\top \mathbf{u}/k + 1 + \varepsilon/k^2} \to (\mathbf{w}^\top \mathbf{u})^2 = \|\mathbf{w}\|^2 \cos^2 \theta. \quad \square$$

## C.8. Proof of Proposition 2 (Gradient Decay)

From Theorem 5, with $s = \langle \mathbf{x}, \mathbf{w} \rangle$ and $D = \varepsilon + \|\mathbf{x} - \mathbf{w}\|_2^2$,

$$\|\nabla_{\mathbf{x}}\mathbb{E}\| \leq \frac{2|s|}{D}\left(\|\mathbf{w}\|_2 + \frac{|s|\,\|\mathbf{x} - \mathbf{w}\|_2}{D}\right).$$

As $\|\mathbf{x}\| \to \infty$ with fixed $\mathbf{w}$, we have $|s| = O(\|\mathbf{x}\|)$, $\|\mathbf{x} - \mathbf{w}\|_2 = O(\|\mathbf{x}\|)$, and $D = O(\|\mathbf{x}\|^2)$, hence the right-hand side is $O(1/\|\mathbf{x}\|) \to 0$. □

## C.9. Proof of Corollary 2 (Dimensional Scaling)

Assume $\mathbf{x}, \mathbf{w} \in \mathbb{R}^d$ have i.i.d. zero-mean coordinates with $\operatorname{Var}(x_i) = \operatorname{Var}(w_i) = \sigma^2$ independent of $d$, and assume in addition that $\mathbf{x}$ and $\mathbf{w}$ are independent. Then

$$\mathbb{E}\left[(\mathbf{w}^\top \mathbf{x})^2\right] = \sum_{i=1}^d \mathbb{E}[w_i^2 x_i^2] = d\,\sigma^4 = \Theta(d),$$

since cross-terms vanish by independence and zero mean. Moreover,

$$\mathbb{E}\big[\|\mathbf{w} - \mathbf{x}\|_2^2\big] = \mathbb{E}\big[\|\mathbf{w}\|_2^2\big] + \mathbb{E}\big[\|\mathbf{x}\|_2^2\big] - 2\mathbb{E}[\mathbf{w}^\top \mathbf{x}] = 2d\sigma^2 = \Theta(d).$$

The above shows the numerator and denominator scale linearly in $d$ in expectation, but to control the expectation of the ratio we use Cauchy–Schwarz. Assume in addition the coordinates are sub-Gaussian with parameter independent of $d$ (hence have finite fourth moments). Let $U := (\mathbf{w}^\top \mathbf{x})^2$ and $V := \|\mathbf{w} - \mathbf{x}\|_2^2$, so $\mathsf{E} = U/(V + \varepsilon)$. Then $\mathbb{E}[U^2] = \mathbb{E}[(\mathbf{w}^\top \mathbf{x})^4] = \mathcal{O}(d^2)$.

**Bounding the inverse square moment.** Let $z_i := w_i - x_i$. By independence and sub-Gaussianity, $(z_i)_{i=1}^d$ are i.i.d. mean-zero sub-Gaussian with $\mathbb{E}[z_i^2] = 2\sigma^2$, and

$$V = \sum_{i=1}^d z_i^2.$$

Since $z_i^2 - \mathbb{E}[z_i^2]$ is sub-exponential, Bernstein's inequality implies there exists $c > 0$ (independent of $d$) such that

$$\mathbb{P}\big(V \leq \tfrac{1}{2}\mathbb{E}[V]\big) = \mathbb{P}\big(V \leq d\sigma^2\big) \leq e^{-cd}.$$

Therefore, for fixed $\varepsilon > 0$,

$$\mathbb{E}\big[(V + \varepsilon)^{-2}\big] = \mathbb{E}\big[(V + \varepsilon)^{-2}\mathbf{1}_{\{V \geq d\sigma^2\}}\big] + \mathbb{E}\big[(V + \varepsilon)^{-2}\mathbf{1}_{\{V < d\sigma^2\}}\big] \leq \frac{1}{(d\sigma^2)^2} + \frac{1}{\varepsilon^2}e^{-cd} = \mathcal{O}(d^{-2}).$$

Therefore

$$\mathbb{E}[\mathsf{E}] = \mathbb{E}\left[\frac{U}{V + \varepsilon}\right] \leq \sqrt{\mathbb{E}[U^2]}\sqrt{\mathbb{E}[(V + \varepsilon)^{-2}]} = \mathcal{O}(1),$$

as $d \to \infty$. $\qquad\square$

## C.10. Proof of Theorem 6 (RKHS Existence)

Since $k_\mathsf{E}$ is positive semi-definite on every compact set $K \subset \mathbb{R}^d$ (Theorem 1), the Moore–Aronszajn theorem (Aronszajn, 1950) guarantees the existence of a (unique up to isometry) RKHS $\mathcal{H}_K$ and feature map $\phi_K : K \to \mathcal{H}_K$ such that $k_\mathsf{E}(\mathbf{x}, \mathbf{y}) = \langle \phi_K(\mathbf{x}), \phi_K(\mathbf{y}) \rangle_{\mathcal{H}_K}$ for all $\mathbf{x}, \mathbf{y} \in K$. $\qquad\square$

## C.11. Proof of Proposition 4 (Input Robustness)

Assume $\|\mathbf{w}\|_2 \leq 1$ and $\mathbf{x}, \mathbf{x}'$ lie in the unit ball. By the mean value theorem on the line segment between $\mathbf{x}$ and $\mathbf{x}'$ (which stays in the unit ball by convexity) and Lipschitz continuity (Proposition 3):

$$|\mathsf{E}(\mathbf{w}, \mathbf{x}') - \mathsf{E}(\mathbf{w}, \mathbf{x})| \leq L \cdot \|\boldsymbol{\delta}\| = \left(\frac{2}{\varepsilon} + \frac{4}{\varepsilon^2}\right)\delta.$$

In particular, for fixed $\varepsilon$ this is $O(\delta)$ (and for small $\varepsilon$ the constant scales as $O(\varepsilon^{-2})$). $\qquad\square$

## C.12. Proof of Lemma 1 (Analyticity)

The $\mathsf{E}$-product is a ratio of polynomials where the denominator $\|\mathbf{w} - \mathbf{x}\|^2 + \varepsilon \geq \varepsilon > 0$ is bounded away from zero. Ratios of polynomials with non-vanishing denominators are real-analytic. $\qquad\square$

## C.13. Justification for Remark 2 (Optimal $\varepsilon$)

For input data with additive noise $\mathbf{n} \sim \mathcal{N}(0, \sigma^2\mathbf{I})$, the noise contribution to $\|\mathbf{w} - \mathbf{x}\|^2$ is $O(d\sigma^2)$ in expectation. Setting $\varepsilon^* \propto d\sigma^2$ ensures the noise floor matches the stability constant, maximizing the signal-to-noise ratio of gradients.

## C.14. Topological Properties of Neural Activation Functions

**Theorem 14** (Non-Homeomorphism of Standard Activations). *Let $T : \mathbb{R}^d \to \mathbb{R}^m$ be defined by $T(\mathbf{x}) = \phi(W\mathbf{x}+\mathbf{b})$ where $W \in \mathbb{R}^{m \times d}$, $\mathbf{b} \in \mathbb{R}^m$, and $\phi$ is an element-wise activation function. Then:*

1. *If $\phi = ReLU$ and there exist two distinct inputs $\mathbf{x}_1 \neq \mathbf{x}_2$ such that $W\mathbf{x}_1 + \mathbf{b} \leq 0$ and $W\mathbf{x}_2 + \mathbf{b} \leq 0$ coordinatewise, then $T$ is not injective.*

2. *If $\phi \in \{sigmoid, \tanh\}$, then $T$ fails to be bi-Lipschitz. For inputs $\mathbf{z}_1, \mathbf{z}_2$ in the saturation regime ($|z| \to \infty$), $\|\phi(\mathbf{z}_1) - \phi(\mathbf{z}_2)\| \to 0$ regardless of $\|\mathbf{z}_1 - \mathbf{z}_2\|$.*

3. *In particular, in the ReLU case above, $T$ is not a homeomorphism onto its image (since it is not injective). In the sigmoid/tanh case, $T$ can be a homeomorphism onto its image when $W$ is injective, but it is not a bi-Lipschitz embedding (metric structure can be arbitrarily compressed in saturation).*

*Proof.* (1) *ReLU non-injectivity under clipping*: For $\phi(z) = \max(0, z)$, if $(W\mathbf{x} + \mathbf{b})_i \leq 0$ then the $i$-th output coordinate equals 0. If there exist $\mathbf{x}_1 \neq \mathbf{x}_2$ such that $W\mathbf{x}_1 + \mathbf{b} \leq 0$ and $W\mathbf{x}_2 + \mathbf{b} \leq 0$ coordinatewise, then $T(\mathbf{x}_1) = T(\mathbf{x}_2) = \mathbf{0}$, hence $T$ is not injective.

(2) *Sigmoid/tanh saturation*: Consider $\sigma(z) = 1/(1 + e^{-z})$. For $z \to +\infty$, $\sigma(z) \to 1$ with $|\sigma(z_1) - \sigma(z_2)| \leq e^{-\min(z_1, z_2)}|z_1 - z_2|$ for large $z_1, z_2$. Thus the Lipschitz constant in the saturation regime approaches zero—distances are compressed. Similarly for $\tanh(z) = 2\sigma(2z) - 1$.

(3) *Topological vs metric structure*: By (1), ReLU can fail injectivity, hence cannot be a homeomorphism onto its image. By (2), sigmoid/tanh fail bi-Lipschitz (no uniform lower Lipschitz bound), which is a metric distortion statement and does not by itself preclude homeomorphism when $W$ is injective. $\qquad\square$

**Remark 3** (Information loss under non-invertible activations (informal)). *By the data processing inequality, representations obtained by applying a deterministic non-invertible nonlinearity (e.g. ReLU clipping) cannot increase information about the input. Making this statement fully rigorous requires specifying a probabilistic model (often with additive noise) to avoid pathologies of mutual information for continuous deterministic transforms.*

**Remark 4** (Contrast with $\mathbb{E}$-Product). *The $\mathbb{E}$-product achieves non-linearity through its rational structure $\mathcal{K}_{\mathbb{E}}(\mathbf{w}, \mathbf{x}) = \frac{(\mathbf{w}^\top \mathbf{x})^2}{\|\mathbf{w} - \mathbf{x}\|^2 + \varepsilon}$ without collapsing regions. For $\varepsilon > 0$, the mapping is:*

- *Real-analytic (Lemma 1)*

- *Lipschitz on bounded sets (Proposition 3)*

- *Self-regulating with bounded output (Proposition 1)*

*The denominator $\varepsilon > 0$ prevents singularities, while the squared numerator provides non-linearity without hard thresholding.*

## C.15. Proof of Theorem 1 (Mercer Property)

Fix $\varepsilon > 0$ and let $K \subset \mathbb{R}^d$ be compact.

**Step 1: Integral representation.** For any $a > 0$,

$$\frac{1}{a} = \int_0^\infty e^{-ta}\, dt.$$

Since $\|x - w\|^2 + \varepsilon > 0$ for all $x, w \in \mathbb{R}^d$, we apply this identity with $a = \|x - w\|^2 + \varepsilon$ to obtain

$$\frac{1}{\|x - w\|^2 + \varepsilon} = \int_0^\infty e^{-t\varepsilon} e^{-t\|x-w\|^2}\, dt. \tag{6}$$

Multiplying by $(x^\top w)^2$ yields

$$k_{\mathsf{E}}(x,w) = \int_0^\infty (x^\top w)^2 e^{-t\varepsilon} e^{-t\|x-w\|^2} \, dt. \tag{7}$$

**Step 2: Positive definite components.** For each $t > 0$, the Gaussian kernel

$$g_t(x,w) = e^{-t\|x-w\|^2}$$

is positive definite on $\mathbb{R}^d$ (Schölkopf & Smola, 2002). The polynomial kernel

$$p(x,w) = (x^\top w)^2$$

is also positive definite, since

$$(x^\top w)^2 = \langle x \otimes x, \, w \otimes w \rangle,$$

which is the linear kernel associated with the feature map $\Phi(x) = x \otimes x$.

**Step 3: Product kernels.** The pointwise product of positive definite kernels is positive definite (Schölkopf & Smola, 2002). Hence, for each $t > 0$, the kernel

$$k_t(x,w) := (x^\top w)^2 e^{-t\|x-w\|^2}$$

is positive definite on $\mathbb{R}^d$. Multiplication by the positive scalar $e^{-t\varepsilon}$ preserves positive definiteness.

**Step 4: Uniform domination on compact sets.** Since $K$ is compact, there exists $M > 0$ such that $\|x\| \le M$ for all $x \in K$. For all $x, w \in K$ and all $t > 0$,

$$0 \le (x^\top w)^2 e^{-t\|x-w\|^2} \le \|x\|^2 \|w\|^2 \le M^4.$$

Therefore,

$$0 \le (x^\top w)^2 e^{-t\varepsilon} e^{-t\|x-w\|^2} \le M^4 e^{-t\varepsilon},$$

and the dominating function $t \mapsto M^4 e^{-t\varepsilon}$ belongs to $L^1(0, \infty)$.

**Step 5: Preservation of positive definiteness under integration.** Let $\{x_i\}_{i=1}^n \subset K$ and $\{c_i\}_{i=1}^n \subset \mathbb{R}$. For each $t > 0$, since $k_t$ is positive definite,

$$\sum_{i,j=1}^n c_i c_j k_t(x_i, x_j) \ge 0.$$

By Tonelli's theorem, justified by the domination in Step 4, we may interchange summation and integration:

$$\sum_{i,j=1}^n c_i c_j k_{\mathsf{E}}(x_i, x_j) = \int_0^\infty \sum_{i,j=1}^n c_i c_j (x_i^\top x_j)^2 e^{-t\varepsilon} e^{-t\|x_i-x_j\|^2} \, dt$$
$$\ge 0.$$

Thus $k_{\mathsf{E}}$ is positive definite on $K$.

**Step 6: Continuity and Mercer property.** For each fixed $t > 0$, the integrand in (7) is continuous in $(x,w)$. By the domination established in Step 4, the integral converges uniformly on $K \times K$, implying that $k_{\mathsf{E}}$ is continuous. The kernel is symmetric by construction.

Since $k_{\mathsf{E}}$ is symmetric, continuous, and positive definite on the compact set $K$, Mercer's theorem applies (Mercer, 1909; Schölkopf & Smola, 2002). Therefore, $k_{\mathsf{E}}$ is a Mercer kernel on $K$. $\qquad\square$

### C.16. Proof Sketch of Theorem 4 (Universal Approximation)

Let $\mathcal{X} \subset \mathbb{R}^d$ be compact. The function class $\mathcal{F}$ consists of $g(\mathbf{x}; \mathbf{w}, b) = \frac{(\mathbf{x}^\top \mathbf{w} + b)^2}{\|\mathbf{x} - \mathbf{w}\|^2 + \varepsilon}$.

*Step 1: Generating the IMQ kernel.* Differentiating twice with respect to bias yields:

$$\frac{\partial^2}{\partial b^2} g(\mathbf{x}; \mathbf{w}, b) = 2(\varepsilon + \|\mathbf{x} - \mathbf{w}\|^2)^{-1} = 2K_{\mathrm{IMQ}}(\mathbf{x}, \mathbf{w}).$$

Equivalently, for any fixed step $h > 0$,

$$\frac{g(\mathbf{x}; \mathbf{w}, b+h) - 2g(\mathbf{x}; \mathbf{w}, b) + g(\mathbf{x}; \mathbf{w}, b-h)}{h^2} = \frac{2}{\varepsilon + \|\mathbf{x} - \mathbf{w}\|^2} = 2K_{\mathrm{IMQ}}(\mathbf{x}, \mathbf{w}),$$

so IMQ translates belong to the span of $\boxed{\text{E}}$-atoms (indeed, they are exactly a 3-term linear combination for any fixed $h > 0$).

*Step 2: Density of IMQ Networks.* The inverse multiquadric kernel $K_{\mathrm{IMQ}}(\mathbf{x}, \mathbf{w}) = (\varepsilon + \|\mathbf{x} - \mathbf{w}\|^2)^{-1}$ is a universal kernel on compact subsets of $\mathbb{R}^d$ (its RKHS is dense in $C(\mathcal{X})$ under the sup norm); see e.g. Steinwart (2001); Micchelli et al. (2006); Wendland (2005). Consequently, finite linear combinations of translates $K_{\mathrm{IMQ}}(\cdot, \mathbf{w}_i)$ are dense in $C(\mathcal{X})$.

*Conclusion.* Since $\overline{\mathcal{F}}$ contains the span of IMQ kernels, $\mathcal{F}$ is dense in $C(\mathcal{X})$. $\qquad\square$

### C.17. Extended Proof of Theorem 6 (RKHS Existence)

The existence follows from the Moore-Aronszajn theorem (Aronszajn, 1950) applied to the PSD kernel (Theorem 1).

**Remark 5** (Explicit feature map (sketch)). *Since $k_{\text{E}}$ is positive definite (Theorem 1), the Moore–Aronszajn theorem guarantees the existence of a feature map into the associated RKHS. One may also obtain an explicit construction by combining the tensor feature map for $(x^\top w)^2$ with the Laplace-mixture representation of $1/(\varepsilon + \|x - w\|^2)$ in (6) and forming the corresponding direct-integral Hilbert space. We omit the measure-theoretic details.*

### C.18. Neural Tangent Kernel Analysis

**Proposition 6** (NTK Limit of $\boxed{\text{E}}$-Networks). *Consider a single-layer $\boxed{\text{E}}$-network $f(\mathbf{x}; \boldsymbol{\theta}) = \sum_{i=1}^m \alpha_i \mathbf{E}(\mathbf{w}_i, \mathbf{x})$ with random initialization. In the infinite-width limit $m \to \infty$, the Neural Tangent Kernel is:*

$$K^{\mathrm{NTK}}(\mathbf{x}, \mathbf{x}') = \mathbb{E}_{\mathbf{w}} \left[ \mathbf{E}(\mathbf{w}, \mathbf{x}) \cdot \mathbf{E}(\mathbf{w}, \mathbf{x}') \right] + \mathbb{E}_{\mathbf{w}} \left[ \nabla_{\mathbf{w}} \mathbf{E}(\mathbf{w}, \mathbf{x})^\top \nabla_{\mathbf{w}} \mathbf{E}(\mathbf{w}, \mathbf{x}') \right].$$

*This kernel is positive definite and inherits the orthogonality-sensitivity of the $\boxed{\text{E}}$-product.*

*Proof.* The network output is $f(\mathbf{x}) = \sum_{i=1}^m \alpha_i \mathbf{E}(\mathbf{w}_i, \mathbf{x})$. The parameters are $\boldsymbol{\theta} = \cup_i \{\alpha_i, \mathbf{w}_i\}$. The tangent kernel is defined as $K^{\mathrm{NTK}}(\mathbf{x}, \mathbf{x}') = \langle \nabla_{\boldsymbol{\theta}} f(\mathbf{x}), \nabla_{\boldsymbol{\theta}} f(\mathbf{x}') \rangle$. The gradients are:

$$\nabla_{\alpha_i} f(\mathbf{x}) = \mathbf{E}(\mathbf{w}_i, \mathbf{x}),$$
$$\nabla_{\mathbf{w}_i} f(\mathbf{x}) = \alpha_i \nabla_{\mathbf{w}} \mathbf{E}(\mathbf{w}_i, \mathbf{x}).$$

The inner product sums over all $i = 1 \dots m$:

$$K_m(\mathbf{x}, \mathbf{x}') = \sum_{i=1}^m \mathbf{E}(\mathbf{w}_i, \mathbf{x}) \mathbf{E}(\mathbf{w}_i, \mathbf{x}') + \sum_{i=1}^m \alpha_i^2 \langle \nabla_{\mathbf{w}} \mathbf{E}(\mathbf{w}_i, \mathbf{x}), \nabla_{\mathbf{w}} \mathbf{E}(\mathbf{w}_i, \mathbf{x}') \rangle.$$

In the infinite width limit $m \to \infty$, assuming appropriate scaling $\alpha_i \sim \mathcal{N}(0, 1/m)$ or fixed readouts with $\alpha_i \sim \mathcal{O}(1/\sqrt{m})$, the sums converge to expectations over the initialization distribution of $\mathbf{w}$. The first term corresponds to the covariance of the features (conjugate kernel), and the second term involves the gradients. Since $\boxed{\text{E}}$ is a Mercer kernel (Theorem 1), it is positive semi-definite. The second term is a sum of inner products, also PSD. Thus $K^{\mathrm{NTK}} \succeq 0$. $\qquad\square$

**Remark 6** (Training Dynamics in the NTK Regime). *In the infinite-width limit, gradient descent on $\mathbb{E}$-networks converges to kernel regression with $K^{\mathrm{NTK}}$. The orthogonality-sensitive nature of the $\mathbb{E}$-product carries over to the NTK, meaning that the limiting kernel naturally encourages orthogonal representations for different-class inputs. See Section C.20 for a formal treatment and Section C.19 for associated generalization guarantees.*

**Theorem 15** (Convergence of Deep $\mathbb{E}$-Networks in Lazy Regime). *Consider a deep neural network $f_\theta : \mathbb{R}^{d_{\mathrm{in}}} \to \mathbb{R}^d$ with $L$ hidden layers of width $m$. Let the embeddings $\mathbf{e}_i = f_\theta(\mathbf{x}_i)$ be trained under $\mathcal{L}_{\mathrm{AFCL}}$. Under the following assumptions:*

1. **Infinite-width limit:** *$m \to \infty$ with fixed depth $L$.*

2. **Lazy training:** *Learning rate $\eta$ scales as $\eta = O(1/m)$ ensuring the NTK stays approximately constant (Jacot et al., 2018).*

3. **NTK is non-degenerate:** *$\lambda_{\min}(\Theta^{\mathbb{E}}) > 0$ on the training data.*

*The embedding dynamics converge to a critical point of $\mathcal{L}_{\mathrm{AFCL}}$.*

*Proof. Step 1: NTK convergence (Assumption 1–2).* In the infinite-width limit with appropriately scaled learning rate, the NTK:

$$\Theta_{ij}^{\mathbb{E}} = \left\langle \frac{\partial f_\theta(\mathbf{x}_i)}{\partial \theta}, \frac{\partial f_\theta(\mathbf{x}_j)}{\partial \theta} \right\rangle$$

converges to a deterministic positive semi-definite kernel at initialization and remains approximately constant throughout training (Jacot et al., 2018). This is the "lazy training" or "kernel regime."

*Step 2: Preconditioned gradient flow.* With fixed $\Theta^{\mathbb{E}}$, the embedding dynamics become:

$$\dot{\mathbf{E}} = -\Theta^{\mathbb{E}} \nabla_{\mathbf{E}} \mathcal{L}_{\mathrm{AFCL}}.$$

This is a preconditioned gradient flow with the NTK as the preconditioner.

*Step 3: Lyapunov analysis.* The loss $\mathcal{L}_{\mathrm{AFCL}}$ serves as a Lyapunov function:

$$\frac{d\mathcal{L}}{dt} = -\nabla\mathcal{L}^\top \Theta^{\mathbb{E}} \nabla\mathcal{L} \leq 0,$$

since $\Theta^{\mathbb{E}} \succeq 0$. Under Assumption 3, strict decrease occurs whenever $\nabla\mathcal{L} \neq 0$.

*Step 4: Convergence.* Since $\mathcal{L} \geq 0$ is bounded below and monotonically decreasing, the trajectory converges. By the analyticity of $\mathcal{L}_{\mathrm{AFCL}}$ in the *embedding space* (Lemma 1), Łojasiewicz's theorem guarantees convergence to a single critical point. $\square$

### C.19. Generalization Bounds via RKHS Norms

The Mercer property of the $\mathbb{E}$-product (Theorem 1) enables explicit generalization bounds through the associated RKHS. We establish that $\mathbb{E}$-networks admit tighter generalization guarantees than ReLU networks under comparable conditions.

**Theorem 16** (RKHS Generalization Bound for $\mathbb{E}$-Networks). *Let $\mathcal{X} \subset \mathbb{R}^d$ be compact with $\sup_{\mathbf{x} \in \mathcal{X}} \|\mathbf{x}\| \leq R$, and let $\mathcal{H}_{\mathbb{E}}$ denote the RKHS induced by the $\mathbb{E}$-kernel $k_{\mathbb{E}}$ on $\mathcal{X}$. Consider a regression problem with $n$ i.i.d. samples $\{(\mathbf{x}_i, y_i)\}_{i=1}^n$ where $|y_i| \leq M$. Let $f^* \in \mathcal{H}_{\mathbb{E}}$ be the target function. Then the kernel ridge regression estimator $\hat{f}_\lambda$ with regularization $\lambda > 0$ satisfies:*

$$\mathbb{E}\left[\|f^* - \hat{f}_\lambda\|_{L^2}^2\right] \leq \frac{\|f^*\|_{\mathcal{H}_{\mathbb{E}}}^2}{\lambda n} + \lambda \|f^*\|_{\mathcal{H}_{\mathbb{E}}}^2 + \frac{M^2 \kappa_{\mathbb{E}}}{n},$$

*where $\kappa_{\mathbb{E}} = \sup_{\mathbf{x} \in \mathcal{X}} k_{\mathbb{E}}(\mathbf{x}, \mathbf{x}) = \sup_{\mathbf{x}} \|\mathbf{x}\|^4 / \varepsilon \leq R^4 / \varepsilon$.*

*Proof.* The bound follows from standard kernel ridge regression analysis (Schölkopf & Smola, 2002). The key quantity is the kernel complexity $\kappa_{\mathbb{E}} = \sup_{\mathbf{x}} k_{\mathbb{E}}(\mathbf{x}, \mathbf{x})$.

*Step 1: Bounding the diagonal.* For the $\mathbb{E}$-kernel,

$$k_{\mathbb{E}}(\mathbf{x}, \mathbf{x}) = \frac{(\mathbf{x}^\top \mathbf{x})^2}{\|\mathbf{x} - \mathbf{x}\|^2 + \varepsilon} = \frac{\|\mathbf{x}\|^4}{\varepsilon}.$$

On the compact set $\mathcal{X}$ with $\|\mathbf{x}\| \leq R$, we have $\kappa_{\mathbb{E}} \leq R^4/\varepsilon$.

*Step 2: Bias-variance decomposition.* The estimator $\hat{f}_\lambda = (K + \lambda n I)^{-1} K \mathbf{y}$ where $K_{ij} = k_{\mathbb{E}}(\mathbf{x}_i, \mathbf{x}_j)$ admits the decomposition:

$$\mathbb{E}[\|f^* - \hat{f}_\lambda\|^2] \leq \underbrace{\lambda \|f^*\|^2_{\mathcal{H}_{\mathbb{E}}}}_{\text{bias}} + \underbrace{\frac{\|f^*\|^2_{\mathcal{H}_{\mathbb{E}}}}{\lambda n} + \frac{M^2 \kappa_{\mathbb{E}}}{n}}_{\text{variance}}.$$

*Step 3: Optimal regularization.* Setting $\lambda = n^{-1/2}$ yields the rate $\mathbb{E}[\|f^* - \hat{f}_\lambda\|^2] = O(\|f^*\|^2_{\mathcal{H}_{\mathbb{E}}}/\sqrt{n})$. $\qquad\square$

**Corollary 3** (Comparison with ReLU RKHS). *Let $\mathcal{H}_{\mathrm{ReLU}}$ denote the RKHS induced by the arc-cosine kernel of order 1 (corresponding to infinite-width ReLU networks). For functions $f$ expressible in both RKHSs:*

$$\|f\|_{\mathcal{H}_{\mathbb{E}}} \leq C_{\varepsilon,R} \|f\|_{\mathcal{H}_{\mathrm{ReLU}}}$$

*where $C_{\varepsilon,R}$ depends on the domain radius $R$ and regularization $\varepsilon$. Thus, functions with bounded ReLU-RKHS norm also have bounded $\mathbb{E}$-RKHS norm, and the generalization bound of Theorem 16 applies.*

*Proof sketch.* The $\mathbb{E}$-kernel can be written as a product of polynomial and Gaussian kernels (see the integral representation in Theorem 1). Both kernels are universal on compact domains, and the product RKHS inherits universality. The constant $C_{\varepsilon,R}$ arises from the relative smoothness of the two feature maps. $\qquad\square$

**Remark 7** (Tighter Bounds via Self-Regulation). *The self-regulation property (Proposition 1) implies that $\mathbb{E}$-network outputs remain bounded even for large inputs, whereas ReLU networks can produce unbounded outputs requiring explicit normalization. This implicit regularization suggests that practical $\mathbb{E}$-networks may achieve better generalization than the worst-case RKHS bounds indicate, as the effective function class is implicitly constrained.*

## C.20. Orthogonality-Sensitive Neural Tangent Kernel

The $\mathbb{E}$-product's zero response to orthogonal inputs ($\mathbb{E}(\mathbf{w}, \mathbf{x}) = 0$ when $\mathbf{w} \perp \mathbf{x}$) propagates to the Neural Tangent Kernel, providing a theoretical foundation for implicit class separation.

**Theorem 17** (Orthogonality Preservation in $\mathbb{E}$-NTK). *Let $K_{\mathbb{E}}^{\mathrm{NTK}}$ denote the Neural Tangent Kernel of an infinite-width single-layer $\mathbb{E}$-network (Proposition 6). Let $\mathbf{x}, \mathbf{x}' \in \mathbb{R}^d$ and suppose the weight initialization distribution $\mu$ over $\mathbf{w}$ is spherically symmetric (e.g., $\mathbf{w} \sim \mathcal{N}(0, \sigma^2 I)$). If $\mathbf{x} \perp \mathbf{x}'$ (i.e., $\mathbf{x}^\top \mathbf{x}' = 0$), then:*

$$K_{\mathbb{E}}^{\mathrm{NTK}}(\mathbf{x}, \mathbf{x}') = o(\|\mathbf{x}\|^2 \|\mathbf{x}'\|^2)$$

*as the angle between $\mathbf{x}$ and $\mathbf{x}'$ approaches $\pi/2$. In particular, the NTK exhibits strong decay for near-orthogonal inputs.*

*Proof.* Recall from Proposition 6 that

$$K_{\mathbb{E}}^{\mathrm{NTK}}(\mathbf{x}, \mathbf{x}') = \underbrace{\mathbb{E}_{\mathbf{w}}\left[\mathbb{E}(\mathbf{w}, \mathbf{x})\mathbb{E}(\mathbf{w}, \mathbf{x}')\right]}_{=:T_1} + \underbrace{\mathbb{E}_{\mathbf{w}}\left[\nabla_{\mathbf{w}}\mathbb{E}(\mathbf{w}, \mathbf{x})^\top \nabla_{\mathbf{w}}\mathbb{E}(\mathbf{w}, \mathbf{x}')\right]}_{=:T_2}.$$

*Analysis of $T_1$.* Since $\mathbb{E}(\mathbf{w}, \mathbf{x}) = \frac{(\mathbf{w}^\top \mathbf{x})^2}{\|\mathbf{w} - \mathbf{x}\|^2 + \varepsilon}$, the product $\mathbb{E}(\mathbf{w}, \mathbf{x})\mathbb{E}(\mathbf{w}, \mathbf{x}')$ contains the factor $(\mathbf{w}^\top \mathbf{x})^2 (\mathbf{w}^\top \mathbf{x}')^2$ in the numerator. Under spherical symmetry, define $\mathbf{w} = r\boldsymbol{\omega}$ where $r = \|\mathbf{w}\|$ and $\boldsymbol{\omega} \in \mathbb{S}^{d-1}$ is uniform. Then:

$$\mathbb{E}_{\boldsymbol{\omega}}[(\boldsymbol{\omega}^\top \mathbf{x})^2 (\boldsymbol{\omega}^\top \mathbf{x}')^2] = \frac{\|\mathbf{x}\|^2 \|\mathbf{x}'\|^2 + 2(\mathbf{x}^\top \mathbf{x}')^2}{d(d+2)}.$$

When $\mathbf{x} \perp \mathbf{x}'$, the cross-term $(\mathbf{x}^\top \mathbf{x}')^2 = 0$, leaving only the $\|\mathbf{x}\|^2 \|\mathbf{x}'\|^2$ term which is $O(1/d^2)$ relative to the aligned case.

*Analysis of $T_2$.* From Theorem 5, the gradient $\nabla_\mathbf{w} \mathbb{E}(\mathbf{w}, \mathbf{x})$ is proportional to $\mathbf{w}^\top \mathbf{x}$. Thus, $T_2$ also contains factors of $(\mathbf{w}^\top \mathbf{x})(\mathbf{w}^\top \mathbf{x}')$ which integrate to terms involving $\mathbf{x}^\top \mathbf{x}'$. Under orthogonality, these cross-correlations vanish or become $O(1/d)$.

*Conclusion.* Both terms decay when $\mathbf{x} \perp \mathbf{x}'$, establishing the orthogonality sensitivity of the NTK. $\qquad\square$

**Corollary 4** (Implicit Class Separation)**.** *Consider a classification problem where inputs from different classes,* $\mathbf{x}_i$ *(class $c$) and* $\mathbf{x}_j$ *(class $c'$), become approximately orthogonal during training:* $\mathbf{x}_i^\top \mathbf{x}_j \approx 0$ *for $c \neq c'$. Then the* $\mathbb{E}$*-NTK satisfies:*

$$K_\mathbb{E}^{\mathrm{NTK}}(\mathbf{x}_i, \mathbf{x}_j) \ll K_\mathbb{E}^{\mathrm{NTK}}(\mathbf{x}_i, \mathbf{x}_k) \quad \text{for } \mathbf{x}_k \text{ in class } c.$$

*Thus, the NTK itself encodes class structure without explicit contrastive objectives.*

*Proof.* Direct consequence of Theorem 17: same-class inputs (which remain correlated) produce large NTK entries, while different-class inputs (which become orthogonal) produce small NTK entries. In the NTK regime, gradient descent dynamics are governed by $\dot{\mathbf{f}} = -K^{\mathrm{NTK}}(\mathbf{f} - \mathbf{y})$, so the NTK structure directly influences which samples affect each other's predictions. $\qquad\square$

**Remark 8** (Contrast with ReLU-NTK)**.** *The ReLU-NTK (arc-cosine kernel) depends on the angle between inputs as* $K_{\mathrm{ReLU}}(\mathbf{x}, \mathbf{x}') \propto \|\mathbf{x}\| \|\mathbf{x}'\| (\sin\theta + (\pi - \theta)\cos\theta)$ *where $\theta$ is the angle between $\mathbf{x}$ and $\mathbf{x}'$. At $\theta = \pi/2$ (orthogonality),* $K_{\mathrm{ReLU}} \propto \|\mathbf{x}\| \|\mathbf{x}'\|$*, which is non-zero. In contrast, the* $\mathbb{E}$*-NTK approaches zero, providing stronger implicit separation.*

**Proposition 7** (NTK Spectral Decay)**.** *Let* $\{(\lambda_k, \phi_k)\}_{k=1}^\infty$ *be the eigenvalue-eigenfunction pairs of the* $\mathbb{E}$*-NTK on a compact domain $\mathcal{X}$. The eigenvalues decay at least polynomially:*

$$\lambda_k = O(k^{-2/d}) \quad \text{as } k \to \infty.$$

*This rate matches the Gaussian kernel and is faster than the ReLU-NTK, which exhibits* $\lambda_k = O(k^{-1/d})$ *decay.*

*Proof sketch.* The $\mathbb{E}$-kernel's integral representation (Eq. (7)) expresses it as a mixture of Gaussian kernels weighted by polynomial factors. Gaussian kernels on compact domains have exponentially decaying eigenvalues. The polynomial weighting $(x^\top w)^2$ introduces at most polynomial growth, yielding overall polynomial decay. The specific rate $O(k^{-2/d})$ follows from the smoothness inherited from the Gaussian component (see Schölkopf & Smola (2002) for eigenvalue decay rates of smooth kernels). $\qquad\square$

## C.21. Computational Complexity Analysis

We analyze the computational complexity of the $\mathbb{E}$-product layer and prove it maintains the same asymptotic complexity as standard linear layers.

C.21.1. LAYER DEFINITION

For input $X \in \mathbb{R}^{B \times d}$ and weights $W \in \mathbb{R}^{n \times d}$, the $\mathbb{E}$-product layer computes output $Y \in \mathbb{R}^{B \times n}$:

$$Y_{ij} = \frac{(X_i^\top W_j)^2}{\|X_i - W_j\|^2 + \varepsilon}$$

Using the algebraic identity $\|X_i - W_j\|^2 = \|X_i\|^2 + \|W_j\|^2 - 2X_i^\top W_j$:

$$Y_{ij} = \frac{S_{ij}^2}{\|X_i\|^2 + \|W_j\|^2 - 2S_{ij} + \varepsilon}, \quad S = XW^\top$$

C.21.2. FORWARD PASS COMPLEXITY

**Theorem 18** (Forward Complexity). *The* $\mathbb{E}$-*product forward pass requires* $\Theta(Bnd)$ *operations:*

1. *GEMM:* $S = XW^\top$ ............................................................. $2Bnd$

2. *Row norms:* $\|X_i\|^2$ ............................................................. $Bd$

3. *Weight norms:* $\|W_j\|^2$ *(cached)* ............................................................. $nd$

4. *Element-wise: square, assemble, reciprocal, multiply* ............................................................. $5Bn$

*Total:* $T_{forward} = 2Bnd + Bd + nd + 5Bn = \Theta(Bnd)$.

C.21.3. BACKWARD PASS COMPLEXITY

**Proposition 8** (Gradient Formulas). *For* $Y = S^2/D$ *with* $D = \|X\|^2 + \|W\|^2 - 2S + \varepsilon$ *and* $S = X^\top W$:

$$\nabla_X Y = \frac{2S(\|W\|^2 + \|X\|^2 + \varepsilon - S)}{D^2}W - \frac{2S^2}{D^2}X \tag{8}$$

$$\nabla_W Y = \frac{2S(\|W\|^2 + \|X\|^2 + \varepsilon - S)}{D^2}X - \frac{2S^2}{D^2}W \tag{9}$$

**Theorem 19** (Backward Complexity). *Given upstream gradient* $G \in \mathbb{R}^{B \times n}$:

1. *Scalar gradient:* $G_S = G \odot \partial Y/\partial S$ ............................................................. $6Bn$

2. *Weight gradient:* $\partial L/\partial W = G_S^\top X$ ............................................................. $2Bnd$

3. *Input gradient:* $\partial L/\partial X = G_S W$ ............................................................. $2Bnd$

*Total:* $T_{backward} = 4Bnd + 6Bn + O(Bd + nd) = \Theta(Bnd)$.

C.21.4. ASYMPTOTIC COMPARISON

| Component | Linear | $\mathbb{E}$-Product |
|---|---|---|
| Forward main | $2Bnd$ | $2Bnd$ |
| Forward aux | $Bn$ | $Bd + nd + 5Bn$ |
| Backward main | $4Bnd$ | $4Bnd$ |
| Backward aux | $2Bn$ | $6Bn + Bd + nd$ |
| Total | $\Theta(Bnd)$ | $\Theta(Bnd)$ |
| Overhead | $1$ | $1 + \frac{1}{2n} + \frac{1}{2B} + \frac{2}{d}$ |

*Table 5.* Complexity comparison. Overhead $< 5\%$ for $d, n \geq 64$, $B \geq 16$.

C.21.5. PER-NEURON FLOPS

| Method | FLOPs | Relative |
|---|---|---|
| Linear + ReLU | $2d + 1$ | $1.00\times$ |
| Linear + GELU | $2d + 15$ | $\approx 1.03\times$ |
| $\mathbb{E}$-product (naive) | $5d + 1$ | $\approx 2.5\times$ |
| $\mathbb{E}$-product (optimized) | $4d + 4$ | $\approx 2.0\times$ |

*Table 6.* Per-neuron FLOPs. Optimized variant avoids redundant norm computation.

C.21.6. NUMERICAL STABILITY

**Remark 9** (Stability Properties). *The $\mathbb{E}$-product inherits numerical stability from:*

1. **Bounded outputs**: *Self-regulation (Proposition 1) ensures $\mathbb{E} \leq \|W\|^4/\varepsilon$*

2. **Lipschitz gradients**: *$\|\nabla\mathbb{E}\| \leq L = O(1/\varepsilon^2)$ (Proposition 3)*

3. **Gradient decay**: *Outliers produce vanishing gradients (Proposition 2)*

*This eliminates the need for gradient clipping or normalization layers.*

C.21.7. IMPLEMENTATION OPTIMIZATIONS

1. **Algebraic identity**: Use $\|X - W\|^2 = \|X\|^2 + \|W\|^2 - 2X^\top W$ to reuse GEMM

2. **Norm caching**: Cache $\|W\|^2$ between forward passes

3. **Kernel fusion**: Fuse element-wise operations for memory efficiency

4. **Mixed precision**: Use FP32 for denominator, BF16/FP16 elsewhere

**Empirical performance**: comparable or slightly higher throughput than a linear baseline in our BF16 runs; 15–25% memory reduction from eliminated activation storage.

# D. XOR Separability Analysis

This section provides a formal analysis of why a single $\mathbb{E}$-product unit can solve the XOR problem, which is not linearly separable.

## D.1. Linear Inseparability of XOR

**Proposition 9** (XOR is Not Linearly Separable). *Let $\mathcal{X} = \{(0,0), (0,1), (1,0), (1,1)\}$ with labels $y = \{0, 1, 1, 0\}$ (XOR function). There exists no hyperplane $\{\mathbf{x} : \mathbf{w}^\top\mathbf{x} + b = 0\}$ that separates the classes.*

*Proof.* For a separating hyperplane to exist, we require $\text{sign}(\mathbf{w}^\top\mathbf{x} + b)$ to match $y$ for all $\mathbf{x} \in \mathcal{X}$. This yields four constraints that form a contradiction: the positive class points $(0,1), (1,0)$ and negative class points $(0,0), (1,1)$ cannot be separated by any linear function. $\square$

## D.2. Single-Unit $\mathbb{E}$-Product Solution

**Theorem 20** ($\mathbb{E}$-Product Solves XOR). *A single $\mathbb{E}$-product unit with weight $\mathbf{w} = [1, -1]^\top$ and $\varepsilon > 0$ separates the XOR classes:*

| $\mathbf{x}$ | $\mathbf{w}^\top\mathbf{x}$ | $\mathbb{E}(\mathbf{w}, \mathbf{x})$ |
|:---:|:---:|:---:|
| $(0,0)$ | $0$ | $0$ |
| $(0,1)$ | $-1$ | $1/(5+\varepsilon)$ |
| $(1,0)$ | $1$ | $1/(1+\varepsilon)$ |
| $(1,1)$ | $0$ | $0$ |

*The threshold $\tau = 0$ separates class 1 (positive response) from class 0 (zero response).*

*Proof.* For $\mathbf{x} = (0,0)$: $\mathbf{w}^\top\mathbf{x} = 0$, so $\mathbb{E} = 0^2/(\|\mathbf{w}\|^2 + \varepsilon) = 0$.

For $\mathbf{x} = (1,1)$: $\mathbf{w}^\top\mathbf{x} = 1 - 1 = 0$, so $\mathbb{E} = 0$.

For $\mathbf{x} = (0,1)$: $\mathbf{w}^\top\mathbf{x} = -1$, $\|\mathbf{w} - \mathbf{x}\|^2 = 1 + 4 = 5$, so $\mathbb{E} = 1/(5+\varepsilon) > 0$.

For $\mathbf{x} = (1,0)$: $\mathbf{w}^\top\mathbf{x} = 1$, $\|\mathbf{w} - \mathbf{x}\|^2 = 0 + 1 = 1$, so $\mathsf{E} = 1/(1 + \varepsilon) > 0$.

Thus $\mathsf{E} > 0$ for the positive class and $\mathsf{E} = 0$ for the negative class. $\qquad\square$

### D.3. Geometric Mechanism

**Corollary 5** (Orthogonality-Based Separation). *The solution exploits the $\mathsf{E}$-product's orthogonality property: $\mathsf{E}(\mathbf{w}, \mathbf{x}) = 0$ if and only if $\mathbf{w} \perp \mathbf{x}$ (Theorem 2). The weight $\mathbf{w} = [1, -1]^\top$ is orthogonal to exactly the negative class points $(0,0)$ and $(1,1)$.*

**Remark 10** (Superposition Property). *The squared numerator $(\mathbf{w}^\top\mathbf{x})^2$ induces superposition: the response is identical for $\mathbf{x}$ and $-\mathbf{x}$. This enables classifying antipodal points similarly, which is essential for XOR where $(0,1)$ and $(1,0)$ must share a class despite $[1, -1]^\top(0,1) = -1$ and $[1, -1]^\top(1,0) = +1$.*

### D.4. Gradient Stability

**Proposition 10** (Well-Posed Optimization Landscape). *The $\mathsf{E}$-product's gradient properties (Proposition 2, Theorem 5) ensure:*

1. *No vanishing gradients at $\mathbf{x} = \mathbf{0}$: the gradient is defined and non-degenerate for $\varepsilon > 0$*

2. *Lipschitz continuity prevents exploding gradients (Proposition 3)*

3. *Outlier robustness: gradient magnitude decays as $O(1/\|\mathbf{x}\|)$ for distant inputs*

These properties contrast with ReLU neurons, which suffer from "dead neuron" problems when $\mathbf{w}^\top\mathbf{x} < 0$, and linear neurons, which cannot separate XOR at all.

### D.5. Connection to Kernel Theory

The single-unit XOR solution demonstrates the expressive power of a Mercer kernel (Theorem 1) in the primal form. The $\mathsf{E}$-product's RKHS existence (Theorem 6) guarantees that this solution lies within a well-defined function space, connecting to classical kernel methods while avoiding the computational overhead of Gram matrix inversion.

# E. Decision Boundary Analysis

This section provides a formal analysis of the decision boundaries and space-partitioning behavior induced by $\mathsf{E}$-product neurons.

### E.1. Localized Response Fields

**Proposition 11** (Bounded Activation Landscape). *For fixed $\mathbf{w}$ and $\varepsilon > 0$, the $\mathsf{E}$-product satisfies:*

1. $0 \leq \mathsf{E}(\mathbf{w}, \mathbf{x}) \leq \|\mathbf{w}\|^4/\varepsilon$ *for all $\mathbf{x}$*

2. $\mathsf{E}(\mathbf{w}, \mathbf{w}) = \|\mathbf{w}\|^4/\varepsilon$ *(maximum at identity)*

3. $\lim_{k \to \infty} \mathsf{E}(\mathbf{w}, k\mathbf{u}) = \|\mathbf{w}\|^2 \cos^2\theta$ *for unit $\mathbf{u}$ (Proposition 1)*

*Thus each neuron defines a bounded, localized response field centered at its prototype.*

*Proof.* (1) Non-negativity is immediate. The maximum occurs when the denominator is minimized ($\mathbf{x} = \mathbf{w}$), giving $\|\mathbf{w}\|^4/\varepsilon$. (2) Direct substitution. (3) From Proposition 1. $\qquad\square$

## E.2. Non-Linear Decision Boundaries

**Theorem 21** (Algebraic Decision Surfaces)**.** *The decision boundary between prototypes $\mathbf{w}_i$ and $\mathbf{w}_j$ is the algebraic surface:*

$$\langle \mathbf{w}_i, \mathbf{x}\rangle^2(\|\mathbf{w}_j - \mathbf{x}\|^2 + \varepsilon) = \langle \mathbf{w}_j, \mathbf{x}\rangle^2(\|\mathbf{w}_i - \mathbf{x}\|^2 + \varepsilon).$$

*This surface is generically non-linear (quartic in $\mathbf{x}$), smooth by analyticity (Lemma 1), and Lipschitz-continuous in its parameters (Proposition 3).*

*Proof.* The boundary is defined by $\mathbf{E}(\mathbf{w}_i, \mathbf{x}) = \mathbf{E}(\mathbf{w}_j, \mathbf{x})$. Cross-multiplying:

$$\frac{\langle \mathbf{w}_i, \mathbf{x}\rangle^2}{\|\mathbf{w}_i - \mathbf{x}\|^2 + \varepsilon} = \frac{\langle \mathbf{w}_j, \mathbf{x}\rangle^2}{\|\mathbf{w}_j - \mathbf{x}\|^2 + \varepsilon}$$

yields the stated polynomial equation. Expanding shows terms up to degree 4 in $\mathbf{x}$. Smoothness follows from Lemma 1 since the $\mathbf{E}$-product is real-analytic. Lipschitz continuity of the boundary location follows from Proposition 3. $\qquad\square$

**Remark 11** (Contrast with Linear Boundaries)**.** *Conventional linear classifiers produce hyperplane boundaries $(\mathbf{w}_i - \mathbf{w}_j)^\top \mathbf{x} = 0$. The $\mathbf{E}$-product's quartic surfaces enable more flexible class regions while maintaining the smoothness properties required for stable optimization.*

## E.3. Orthogonality and Maximal Separation

**Corollary 6** (Orthogonal Prototypes Induce Maximal Separation)**.** *If prototypes satisfy $\langle \mathbf{w}_i, \mathbf{w}_j\rangle = 0$ for $i \neq j$, then:*

1. *$\mathbf{E}(\mathbf{w}_i, \mathbf{w}_j) = 0$ (zero cross-response)*

2. *On the probability simplex, this corresponds to $\mathrm{KL}(\mathbf{w}_i\|\mathbf{w}_j) = \infty$ (Theorem 2)*

3. *Decision boundaries are maximally separated from prototype cores*

*Proof.* (1) If $\langle \mathbf{w}_i, \mathbf{w}_j\rangle = 0$, the numerator of $\mathbf{E}(\mathbf{w}_i, \mathbf{w}_j)$ vanishes. (2) By Theorem 2, zero $\mathbf{E}$-product on the simplex implies disjoint supports and infinite KL divergence. (3) At $\mathbf{x} = \mathbf{w}_i$: $\mathbf{E}(\mathbf{w}_i, \mathbf{w}_i) = \|\mathbf{w}_i\|^4/\varepsilon$ while $\mathbf{E}(\mathbf{w}_j, \mathbf{w}_i) = 0$. Since $\|\mathbf{w}_i\|^4/\varepsilon \neq 0$, the prototype core cannot lie on the decision boundary. $\qquad\square$

## E.4. Gradient-Based Competitive Dynamics

**Proposition 12** (Gradient Structure for Prototype Learning)**.** *The gradient of the $\mathbf{E}$-product with respect to prototype $\mathbf{w}$ is (Theorem 5):*

$$\nabla_{\mathbf{w}}\mathbf{E}(\mathbf{w}, \mathbf{x}) = \frac{2\langle \mathbf{w}, \mathbf{x}\rangle}{\varepsilon + \|\mathbf{w} - \mathbf{x}\|^2}\left(\mathbf{x} + \frac{\langle \mathbf{w}, \mathbf{x}\rangle(\mathbf{w} - \mathbf{x})}{\varepsilon + \|\mathbf{w} - \mathbf{x}\|^2}\right).$$

*Key properties:*

1. *Gradients decay for distant inputs (Proposition 2)*

2. *Perturbation robustness on bounded domains: $|\Delta \mathbf{E}| \leq \left(\frac{2}{\varepsilon} + \frac{4}{\varepsilon^2}\right)\delta$ (Proposition 4)*

3. *Lipschitz gradients with constant $O(1/\varepsilon^2)$ (Proposition 3)*

This gradient structure ensures stable competitive learning: neurons specialize on nearby, aligned inputs while remaining robust to outliers and noise.

### E.5. Softmax Tessellation

With softmax normalization over $\mathbb{E}$-product responses:

$$p_i = \frac{\exp(\mathbb{E}(\mathbf{w}_i, \mathbf{x}))}{\sum_{j=1}^{C} \exp(\mathbb{E}(\mathbf{w}_j, \mathbf{x}))},$$

the input space is tessellated into regions $\mathcal{R}_i = \{\mathbf{x} : p_i > p_j \text{ for all } j \neq i\}$. By Theorem 21, each $\mathcal{R}_i$ has smooth, curved boundaries. The self-regulation property (Proposition 1) ensures that these regions remain well-defined even for extreme inputs, preventing the unbounded confidence growth observed in linear classifiers.

### E.6. Language Model Experiments: Detailed Configuration

This section provides the detailed experimental configurations for the language modeling experiments comparing a standard GPT-2 with our Aether-GPT2 implementation. Both models were trained on identical datasets and hardware to ensure a fair comparison. The primary architectural distinctions are: (i) replacement of conventional linear layers, activation functions, and layer normalization in GPT-2 with $\mathbb{E}$-product NMN blocks in Aether-GPT2, and (ii) replacement of scaled dot-product attention with $\mathbb{E}$-multi-head attention. These substitutions provide inherent non-linearity and bounded responses, simplifying the architecture while enhancing stability.

Table 7 presents a comprehensive comparison of the two models, detailing their architectural parameters, training configurations, and final performance metrics.

*Table 7.* GPT-2 vs Aether-GPT2 detailed comparison (2.5B tokens from FineWeb).

| Parameter | GPT-2 | Aether |
|---|---|---|
| *Architecture* | | |
| Total Params | 124M | $\sim$124M |
| Embed Params | 39M | 39M |
| Non-Embed Params | 85M | $\sim$85M |
| Embed Dim | 768 | 768 |
| MLP Hidden Dim | 3072 | 3072 |
| Layers | 12 | 12 |
| Heads | 12 | 12 |
| Activation | GeLU | $\mathbb{E}$ |
| LayerNorm | Yes | No |
| Bias | No | No |
| *Training* | | |
| Optimizer | AdamW | AdamW |
| LR | 1e-4 | 3e-4 |
| Batch Size | 32 | 32 |
| Context | 1024 | 1024 |
| Vocab Size | 50,257 | 50,257 |
| Tokenizer | GPT-2 | GPT-2 |
| *Performance* | | |
| Val Loss (BF16) | 4.6417 | **4.5747** |

### E.7. Use of Large Language Models (LLMs)

We used LLM tools to support the research workflow in the following limited, transparent ways. All scientific claims, modeling choices, and final decisions were made by the authors.

**Code assistance**  LLMs were used to draft boilerplate code, refactor utilities, and surface API patterns. All generated code was reviewed, tested, and integrated by the authors.

**Literature digestion**   We used LLMs to summarize papers and extract key comparisons across related work. Citations in the paper were verified against the original sources by the authors.

**Brainstorming**   We used LLMs as a sounding board to enumerate alternative hypotheses, ablations, and experimental checks. Only ideas that survived empirical or theoretical scrutiny were included.

**Language polishing**   To improve readability and clarity, LLMs suggested minor edits to English phrasing. Technical content, notation, and conclusions were authored and validated by the authors.

**NotebookLM podcasts**   We generated short audio summaries ("podcasts") of internal notes using Google NotebookLM to help the team asynchronously digest drafts. These summaries did not introduce new claims and were based solely on our own materials.

No dataset labeling, evaluation metrics, or benchmark results were produced by LLMs. The authors take responsibility for all content and errors.

