# OpenReview forum: "No More DeLuLu: Physics-Inspired Kernel Networks for Geometrically-Grounded Neural Computation"
_ICML.cc/2026/Conference — Submitted to ICML 2026_

### Official Review · Reviewer_BukK · 2026-03-07

**Soundness:** 2
**Presentation:** 2
**Significance:** 2
**Originality:** 3
**Overall Recommendation:** 2
**Confidence:** 4

**Summary:**

This paper proposes a new activation-free neural operator, the **ⵟ-product**, defined as a ratio between a squared alignment term and a distance-based term. Building on this operator, the paper introduces **Neural Matter Networks (NMNs)** and argues that the resulting computation can be interpreted through kernel, RKHS, and information-geometric perspectives. The paper further develops activation-free replacements for standard neural network components, including MLP blocks, attention-style mechanisms, and normalization-free transformer-style architectures.

Empirically, the paper evaluates the proposed operator on several settings, including small-scale image classification, language modeling, and in-context regression. The main claim is that geometry-based, activation-free computation built from the ⵟ-product can serve as a viable alternative to conventional linear-plus-activation network designs, while also offering a unifying theoretical perspective on representation and interaction.

**Compliance With Llm Reviewing Policy:**

Affirmed.

**Key Questions For Authors:**

1. **Can you isolate which architectural changes are actually responsible for the GPT-2 result in Table 3?**
   As presented, Aether-GPT2 differs from the GPT-2 baseline in several ways at once: the MLP block is changed, attention is changed, LayerNorm is removed, and the detailed training configuration in **Appendix E.6 / Table 7** also uses a different learning rate. A cleaner attribution study would make the main empirical claim much easier to evaluate. In particular, I would like to know whether the gain persists when changing only one component at a time.

2. **What should readers take as the paper’s main validated claim today: a new operator with interesting geometric properties, or a competitive activation-free replacement for standard deep-learning blocks?**
   Right now the paper makes both claims, but the evidence supporting them is uneven. A more precise statement of scope would help. For example, if the main claim is primarily theoretical or conceptual, I would assess the paper differently than if the main claim is that NMNs are already a strong empirical alternative to standard architectures.

3. **Can you provide stronger empirical evidence that the operator yields robust gains beyond the current small or mixed-result settings?**
   The MNIST result is very limited, and the extreme-classification result in **Table 2** is mixed across metrics. The language-modeling result is the most promising part, but it is still relatively modest in size. Any additional evidence—especially on a cleaner benchmark or with stronger ablations—would help clarify whether this is a generally useful modeling primitive or an interesting but still preliminary idea.

4. **Which theoretical results do you view as essential for the empirical story, and which are included mainly as broader context?**
   The paper contains a large amount of theory—Mercer structure, RKHS, universal approximation, NTK analysis, information geometry, and generalization bounds—but not all of these seem equally central to the experiments. A sharper guide to which results are actually needed to justify the practical modeling claims would improve the paper substantially.

5. **Do you have a more direct explanation for why removing LayerNorm appears beneficial for Aether-GPT2 in Table 4?**
   This is one of the more interesting observations in the paper, but the current explanation remains somewhat high-level. A more concrete account—geometric, optimization-related, or statistical—would help connect the operator design to the architectural behavior more convincingly.

**Limitations:**

No.

The paper would benefit from a more explicit limitations discussion. In its current form, the manuscript is quite ambitious in both theory and scope, but the practical limitations are not stated clearly enough.

A stronger limitations paragraph should address at least the following points:

1. **Limited empirical validation relative to the breadth of the claims.**
   The paper makes broad claims about activation-free computation and principled alternatives to conventional architectures, but the empirical evidence is still limited and mixed: the MNIST result is weak, the extreme-classification result is not uniformly better across metrics, and the GPT-2 result is promising but modest and confounded by multiple simultaneous architectural changes.

2. **Confounding in the architectural comparisons.**
   Especially in the language-modeling setting, it is difficult to isolate the effect of the proposed operator from other simultaneous changes such as attention replacement, removal of LayerNorm, and different optimization settings. The paper should acknowledge more directly that the current experiments do not yet cleanly identify the source of the observed gains.

3. **Gap between theory and demonstrated practical value.**
   The paper includes substantial theory, but not all of it is tightly connected to the empirical results. It would help to state explicitly that some of the theoretical material provides broader perspective rather than directly validating the practical modeling claims.

4. **Potential societal impact.**
   I do not see a major paper-specific negative societal-impact concern here. However, since the paper discusses replacing standard neural components in large models, the authors could briefly note that improved architectural simplicity or geometric interpretability does not by itself imply safer or more reliable deployment, and that broader downstream claims would require substantially stronger empirical validation.

Overall, the paper would be stronger with a short and explicit limitations section that narrows the scope of the validated claims and distinguishes clearly between demonstrated results and broader ambition.

**Strengths And Weaknesses:**

**Strengths.**
The paper proposes a clear and distinctive operator, the **ⵟ-product**, and builds a fairly broad conceptual framework around it. The idea of combining a squared alignment term with a distance-based denominator into a single activation-free operator is at least novel at the level of architectural design and mathematical framing. The paper also attempts to connect this operator to multiple viewpoints, including Mercer kernels, RKHS structure, NTK analysis, and information geometry.

A second strength is ambition. The paper does not stop at a toy neuron proposal, but tries to instantiate the operator in several settings: small-scale classification, extreme classification, language modeling, and in-context regression. That breadth gives the paper a potentially interesting unifying perspective, and the language-modeling experiment is at least directionally encouraging. In particular, Table 3 reports a validation-loss improvement from **4.6417 to 4.5747** on a 124M-parameter GPT-2-style setup trained on 2.5B FineWeb tokens.

The paper is also reasonably clear at a high level. The main object being proposed is easy to identify, the core intuition is memorable, and the manuscript is organized into theory, architectural instantiations, and experiments in a way that is not hard to follow. The language-modeling ablation on LayerNorm incompatibility is also a useful attempt to connect the operator’s geometry to an architectural design choice.

**Weaknesses.**
My main concern is **soundness**: the empirical evidence is not strong enough to support the scale of the paper’s claims. The introduction and conclusion frame the method as a principled alternative to conventional neural architectures, but the experimental results are much more mixed and limited. On **MNIST**, the paper itself says the NMN classifier only matches or slightly exceeds a linear baseline, which is consistent with the abstract’s wording that it “matches linear baselines.” Given the ambition of the theoretical framing, this is a weak primary vision result.

The language-modeling result is more interesting, but it is still not fully convincing as currently presented. In **Table 3**, the reported validation loss improves from **4.6417** for GPT-2 to **4.5747** for Aether-GPT2, which is a modest **1.45%** relative improvement. At the same time, the comparison changes several ingredients at once: the MLP block is replaced, attention is replaced, LayerNorm is removed, and the learning rate differs between the detailed configurations (**1e-4** for GPT-2 vs **3e-4** for Aether in Appendix E.6 / Table 7). This makes attribution difficult. It is not yet clear whether the gain comes from the ⵟ-product itself, the normalization removal, the altered optimization setting, or their combination.

A related weakness is that the paper frequently presents very broad theoretical claims without equally strong empirical validation. The manuscript emphasizes Mercer-kernel structure, RKHS existence, universal approximation, NTK analysis, and generalization bounds, but many of these results are either standard consequences once positive definiteness is established or are only loosely connected to the main experimental claims. For example, the generalization result in **Theorem 16** is explicitly described as following from “standard kernel ridge regression analysis,” and the universal-approximation argument in **Appendix C.16** relies on recovering IMQ kernels through a bias-differentiation construction. These may be mathematically interesting, but they do not by themselves justify the stronger architectural claims made in the main text.

On **significance**, I found the paper weaker than its presentation suggests. The extreme-classification result in **Table 2** is mixed rather than clearly favorable: the ⵟ-product improves **P@1/P@3/P@5** over the inner-product baseline, but the baseline is still better on **PSP@3** and **PSP@5**, which complicates the message. Likewise, the GPT-2 result is directionally positive but not yet strong enough, in my view, to support the broader claim that NMNs are a compelling alternative to standard transformer components.

I also have concerns about **presentation and claim calibration**. The paper often states conclusions at a level of confidence that the experiments do not yet earn. For example, the conclusion says the work offers a “principled path toward simpler, more interpretable architectures,” and the abstract says the framework “establishes NMNs as a principled alternative to conventional neural architectures.” Given that the main empirical support consists of a weak MNIST result, a mixed extreme-classification result, and a modest GPT-2 validation-loss improvement under a multi-factor architectural change, this framing feels overstated.

Finally, while the paper is certainly **original**, I am less convinced by its practical contribution at the current stage. The most valuable part of the work may be the operator itself and its geometric interpretation, but the manuscript tries to claim too many things at once: new architecture, new theory, better stability, implicit normalization, kernel-method benefits, improved calibration, and transformer replacement. In my view, the paper would be stronger if it narrowed its central contribution and supported it more cleanly with targeted evidence. Overall, I see a creative idea here, but not yet a sufficiently validated ICML paper.

---

> ### Author Rebuttal · Authors · 2026-03-31
>
> We thank the reviewer for identifying what we believe is the central issue: the original manuscript mixed an operator-level claim with a much broader architecture-level claim. We agree this weakened the paper. Our clarified position in rebuttal is narrower: the main supported contribution is a new kernel/operator with clear theoretical structure and early evidence that Yat-based blocks can train stably at 124M scale, not a claim that NMNs already outperform standard architectures. All references to manuscript changes below refer to edits we would make in the camera-ready version.
>
> **Factorized ablation (new during rebuttal).** To address the attribution concern, we ran 5 arms at 124M scale (768d/12h/12L/3072ff, FineWeb-Edu, BF16, TPU v6e-8, 5K steps = 2.6B tokens). The baseline has 3 initialization seeds (data order is fixed via HuggingFace streaming; variance reflects initialization sensitivity only). Other arms are single-seed due to TPU quota constraints.
>
> | Configuration | MLP | Attn | Norm | Val Loss | tok/s |
> |---|---|---|---|---|---|
> | **GPT-2 baseline (A1, 3 seeds)** | GELU | Std | Pre-RMS | **4.3527 ± 0.0041** | 707K |
> | + Yat-MLP only (A2) | YatNMN | Std | Pre-RMS | 4.6942 | 683K |
> | + Yat-Attn only (A3) | GELU | YAT | Pre-RMS | 4.5262 | 523K |
> | + Both + Pre-RMS (A5) | YatNMN | YAT | Pre-RMS | 4.5778 | 484K |
> | + Both + Post-LN (A11) | YatNMN | YAT | Post-LN | 4.5309 | 395K |
>
> Both Yat components increase val loss vs the baseline (MLP: +0.34, attention: +0.17). The GELU baseline remains stronger at this horizon. We do not claim the Yat-product outperforms standard activations.
>
> **Main validated claim (Q2).** Our clarified claim is: a new operator with interesting geometric/kernel properties and evidence of stable operation at transformer scale. We do not claim that NMNs are already a validated replacement for standard architectures at scale. Each NMN unit computes a genuine kernel evaluation K(x,w) between input and prototype in the same ambient space, giving it a kernel-theoretic interpretation at the unit level that standard activations do not support as directly. We are careful to note that the full trained network remains a learned parametric model; classical RKHS guarantees transfer only in regime-specific settings.
>
> **Essential vs. context theory (Q4).** The foundational spine is PSD/Mercer, RKHS existence, and the universal approximation result (via IMQ reduction). Regularity results (self-regularization, dimensional scaling, gradient identity) are architecture-motivating. One practically relevant result is the explicit gradient identity, which implies a minimal residual structure for backward computation; the observed rematerialization pattern on TPU is consistent with this prediction. NTK and generalization bounds are regime-specific context, explicitly labeled as such.
>
> **LayerNorm (Q5).** We withdraw the incompatibility claim entirely. The divergence was due to a numerical precision bug (BF16 overflow), not a geometric incompatibility. With a selective FP32 upcast, LayerNorm converges normally. A11 (Post-LayerNorm, val 4.5309) is the best Yat configuration.
>
> **Extreme classification mixed (W2).** We acknowledge this honestly. Propensity-scored metrics upweight rare labels, where the inner-product baseline may benefit from its unbounded dynamic range. This is a tradeoff, not a clear win.
>
> **Computational cost.** The Yat-product is ~4d+4 FLOPs/neuron, comparable to SwiGLU (4d+1) but with roughly 50% fewer parameters (d+1 vs 2d+1). Peak HBM matches GELU at equal parameter count; SwiGLU/GeGLU use 1.33× more. Throughput: YatNMN MLP swap is 3.5% slower than baseline; Yat-attention is ~26% slower.
>
> **Claim calibration.** The abstract and conclusion would frame the empirical result as stable training with a 4–8% gap, not as superiority. We agree the original manuscript lacked an explicit limitations section; we would add one that distinguishes demonstrated results from broader ambition.

---

> > ### Author Rebuttal · Reviewer_BukK · 2026-04-01
> >
> > Thank you for the rebuttal. I appreciate the authors’ candor, and the clarification is helpful. However, my main concerns remain unresolved. In fact, the rebuttal substantially narrows the paper’s supported claim: the main validated contribution is now presented as a new operator with interesting geometric/kernel structure and early evidence of stable training at transformer scale, rather than a competitive architecture-level alternative to standard designs. This is a much narrower claim than the one made in the current manuscript.
> >
> > More importantly, the new factorized ablation does not support the paper’s strongest empirical message. In the 124M transformer study, the GPT-2 baseline remains stronger than the Yat-based variants at the reported training horizon, so the rebuttal does not establish that the proposed blocks are presently a compelling replacement for standard transformer components. The withdrawal of the LayerNorm incompatibility claim, due to a numerical precision bug, also weakens one of the paper’s original architectural conclusions.
> >
> > I view this as a core issue rather than something that can be resolved in a short rebuttal. The rebuttal improves claim calibration, but it also makes clear that the current manuscript overstates what is empirically supported. My overall assessment therefore remains unchanged.

---

### Official Review · Reviewer_purG · 2026-03-09

**Soundness:** 2
**Presentation:** 1
**Significance:** 2
**Originality:** 2
**Overall Recommendation:** 2
**Confidence:** 5

**Summary:**

The authors introduce the YAT-product, a novel kernel function. They prove several mathematical properties of this product, such as being a Mercer kernel, and propose a corresponding "Neural Matter Network" (NMN) paradigm adapting standard neural networks by replacing the inner product and nonlinearity of a single hidden layer neuron with this product. They prove a universal approximation property of NMNs, derive extensions to CNNs and transformers, and evaluate NMNs on MNIST and a language modelling task.

**Compliance With Llm Reviewing Policy:**

Affirmed.

**Final Justification:**

My original stance was that, while the idea and UAT of the paper were interesting, the empirical evidence, presentation, and other theoretical justification just wasn't there. After the rebuttal, I was pleased to see the scientific quality of the results improve, but these experiments revealed that, under general problem settings, the proposal significantly underperforms standard baselines. As such, the paper is below the acceptance bar in its current form.

**Key Questions For Authors:**

- One of my main problems with the paper is empirical evaluation. If the authors were able to reproduce the existing experiments with a) proper error bars and b) diverse baselines as discussed above (at least three more would be sufficient - a ReLU network, a normalization-free transformer (where applicable), and an RBF- or some other kernel-based network), then I may be inclined to improve my score.
- Currently, justification of the existence of a lazy regime at all is not provided in the paper. If the authors were able to prove this, for example by demonstrating expressibility of NMNs in the Tensor Programs Framework (Yang 2020), then that would improve the significance of the theory surrounding this.
- It is not clear to me what the advantages of the derived sign-flip robustness and orthogonality sensitivity (both inherent to the YAT product and inherited by the NTK) are, especially relative to the mentioned arc-cosine kernels?
- A big part of the paper's motivation is the "geometrical-grounding" of the YAT product and NMNs, but to me this seems almost arbitrary. Why is the geometry underpinning this more valid for neural computation than the geometry underpinning any other kinds of kernels or NNs?
- I don't see why Layer Normalization has to be intrinsically incompatible with NMNs? For example, by making weight vectors arbitrarily large relative to the unit vector $x$ (such that $\lVert x-w\rVert^2$ is approximately constant) and $\varepsilon$ sufficiently small the YAT product acts like a simple squared inner product between $x$ and a unit vector with the same direction as $w$, so is it the lack of elementwise-nonlinearity causing the issues here? In any case, I think this failure mode is an interesting phenomenon and should be discussed more.

Greg Yang. Tensor programs ii: Neural tangent kernel for any architecture. arXiv preprint320 arXiv:2006.14548, 2020

**Limitations:**

No - missing impact statement

**Strengths And Weaknesses:**

# Strengths #
- Soundness: The theoretical results are for the most part correct and sufficiently proven. Most importantly, this statement applies to the universal approximation result.
- Presentation: The figures are generally well-made and informative.
- Significance: The YAT-product is conceptually interesting.
- Originality: The YAT-product is, to the best of the reviewer's knowledge, a novel kernel.

# Weaknesses #
- Soundness: The empirical evidence supporting use of NMNs is severely lacking. On the included experiments, no error bars are provided which puts the statistical significance of results in to question, especially given that the quoted improvements are on the order of a couple of percentage points. The chosen baseline for MNIST is also not a remotely fair match; comparing a network with inherent nonlinear input-output relationships and universal approximation capabilities (NMN) to a linear model without universal approximation is artificial and misleading, and given this discrepancy in capability it is surprising that such a small improvement in results is obtained. Rigorous evaluation should include baselines using a variety of activation functions and normalisation schemes, and ideally also include other variants on kernelized networks (for example replacing the YAT product with a standard RBF kernel, which would also trivially admit many of the important properties derived in the paper). Given the emphasis on the claimed normalization properties of NMNs, it would also be prudent to evaluate against other normalization-free models in both experiments, such as Jiachen et al. 2025. Furthermore, the range of experiments provided is insufficient; I would like to see a much wider array of empirical evaluation settings, for example on ImageNet as a more difficult image processing benchmark with many established comparable baselines. Finally, the paper discusses theory pertaining to the NTK / lazy-training regime, but uses the results of Jacot et al. 2018 without justification that the Hessian indeed decays to 0 in the infinite width limit in NMNs (see the later question for further discussion).
- Presentation: The submission is poorly written and structured in general. An abundance of theory is presented in the main body, and yet the vast majority of this is completely irrelevant to what I percieve to be the main point of the paper, being NMNs. For example, Theorems 2 and 3 are meaningless without further contextualization, which distracts from the rest of the paper. More could be done to emphasize the placement in the wider field of kernelized neural networks, which would lend more support to the inclusion of all these theorems. Instead, the paper mainly aims to place NMNs in the context of normalization, which I think is a misunderstanding of what normalization actually does - for example, it seems possible for the norm of hidden layers to become arbitrarily large or small in this formulation (for wide networks at least) which can only control per-neuron magnitudes. Instead the YAT product behaves much more like a tanh linearity, which only has correspondence to Layer Normalization in an emergent property sense (for example, see Jiachen et al. 2025, Ziomek et al. 2025). In general, the presentation of the significance of the contributions by the paper is overly strong and dismissive of well-established results from the literature. In particular, the portion of the title "No More DeLuLu", which to my understanding implies that the YAT product is intended to replace the use of ReLU, GELU etc nonlinearities, is not at all substantiated by evidence that NMNs outperform NNs using such activation functions, which themselves have well-known universal approximation capabilities etc. Moreover, the assertion that ReLU networks discard information through thresholding is misleading (for example, see Elhage et al. 2022, which suggests this thresholding is acts as a filtering mechanism) - indeed, the YAT product is also guilty of this sense of "information loss" by Propositions 1 and 2, which imply a loss of magnitude information for some inputs which, by the authors own assertion via the failure of NMNs upon inclusion of Layer Norm operations, is important for the function of NMNs. Finally, the latex formatting of the paper is also quite poor, with Table 3 intersecting the text of the adjacent column, many lines consisting of single words, incorrectly formatted quotation marks, and a missing Impact Statement.
- Significance: The paper essentially just introduces a new kernel function for use in neural networks, supposedly with principled geometric backing. I do not at present believe the paper sufficiently justifies this geometric backing (see the below question), and given the lack of sufficient evidence for the efficacy of this kernel, the advances in the field as a result of this paper are incremental.
- Originality: The idea of using kernel operations in neural networks is not itself new, and no new insights into the field are provided by this paper. The novelty again comes purely from the introduction of the YAT product, which is incremental without proper empirical backing.

Zhu, Jiachen, et al. "Transformers without normalization." Proceedings of the computer vision and pattern recognition conference. 2025.
Ziomek, Juliusz, George Whittle, and Michael A. Osborne. "Just One Layer Norm Guarantees Stable Extrapolation." The Thirty-ninth Annual Conference on Neural Information Processing Systems. 2025.
Elhage, N., Hume, T., Olsson, C., Schiefer, N., Henighan, T., Kravec, S., Hatfield-Dodds, Z., Lasenby, R., Drain, D., Chen, C., Grosse, R., McCandlish, S., Kaplan, J., Amodei, D., Wattenberg, M., & Olah, C. (2022). Toy models of superposition. Transformer Circuits. https://transformer-circuits.pub/2022/toy_model/index.html

---

> ### Author Rebuttal · Authors · 2026-03-31
>
> We thank the reviewer for the detailed scrutiny. We agree the original empirical evaluation and presentation did not adequately support the paper's broadest claims. All manuscript changes below refer to edits for the camera-ready version.
>
> ### Empirical evaluation
>
> During rebuttal, we ran a 5-arm factorized ablation at 124M scale (768d/12h/12L, FineWeb-Edu, BF16, TPU v6e-8, 5K steps = 2.6B tokens). Baseline has 3 init seeds; others single-seed (TPU quota).
>
> | Config | MLP | Attn | Norm | Val Loss | tok/s |
> |---|---|---|---|---|---|
> | **Baseline (A1, 3s)** | GELU | Std | Pre-RMS | **4.3527±0.0041** | 707K |
> | +Yat-MLP (A2) | YatNMN | Std | Pre-RMS | 4.6942 | 683K |
> | +Yat-Attn (A3) | GELU | YAT | Pre-RMS | 4.5262 | 523K |
> | +Both (A5) | YatNMN | YAT | Pre-RMS | 4.5778 | 484K |
> | +Both+PostLN (A11) | YatNMN | YAT | Post-LN | 4.5309 | 395K |
>
> Both components increase val loss vs baseline (MLP:+0.34, attn:+0.17). GELU remains stronger. We do not claim Yat outperforms standard activations.
>
> We were unable to complete the three requested baselines (ReLU, norm-free transformer, RBF network) within the rebuttal period due to TPU quota. MNIST is reframed as a prototype dynamics study, not a competitive benchmark.
>
> ### Q1: Lazy regime
>
> We agree the original manuscript did not justify the existence of a lazy regime for NMNs. Our NTK analysis is an infinite-width characterization, not a claim that NMNs operate in the lazy regime during finite-width training. Extending Tensor Programs (Yang, 2020) to NMNs is an open question. NTK/generalization results would be labeled as regime-specific context.
>
> ### Q2: Sign-flip and orthogonality vs arc-cosine kernels
>
> We agree the original paper did not explain clearly enough why these properties matter in practice. **Sign-flip:** motivated by superposition settings (Elhage et al., 2022) where features share neurons with opposite signs. Flipping all 10 MNIST prototypes: Yat retains 87.87% accuracy vs linear's 0.01%. Empirical observation at small scale, not a proven transformer-scale advantage.
>
> **Orthogonality:** at the unit level, ReLU/arc-cosine kernels assign near-zero for theta>=pi/2, collapsing anti-aligned and orthogonal inputs. Yat gives nonzero response to anti-aligned inputs. We present this as a geometric property, not a proven downstream benefit.
>
> **vs arc-cosine:** arc-cosine kernels characterize infinite-width ReLU networks in the dual regime, not used as primal computation primitives. The Yat-product is used directly as the layer computation. Structural difference, not categorical superiority.
>
> ### Q3: Why this geometry?
>
> The original framing overstated uniqueness. The Yat-product has distinctive properties (alignment+proximity encoding, sign-flip invariance, bounded far-field) but is not the uniquely valid geometry. Physics language is intuition; the backbone is kernel-theoretic. Other kernels have their own strengths.
>
> ### Q4: LayerNorm
>
> The reviewer was correct. We withdraw the claim entirely. The divergence was a BF16 precision bug (squaring overflows 7-bit mantissa), not geometric incompatibility. With FP32 upcast for the score path, LayerNorm converges normally. A11 (Post-LN, val 4.5309) is the best Yat config.
>
> ### Presentation
>
> We will organize theory into three tiers: (1) foundational spine (PSD/Mercer, RKHS, UAT), (2) regularity (self-regularization, scaling, gradient identity), (3) regime-specific context (NTK, bounds). The normalization comparison is removed; our strongest result uses Post-LN. Formatting issues would be fixed.
>
> On tanh: the analogy does not hold beyond bounded outputs. tanh has no RKHS/PSD structure. The Yat limit ||w||²cos²θ still encodes angle; tanh saturates to ±1 regardless.
>
> On Props 1-2: K=0 at orthogonality differs from ReLU's zero. ReLU maps the entire half-space to zero (many-to-one); Yat gives K=0 only at exact orthogonality.
>
> ### Significance
>
> The contribution is the specific kernel, its proof strategy (IMQ reduction; we are not aware of prior universal-approximation proofs for this operator using this reduction), and evidence that it can train stably at 124M scale, including under standard normalization.
>
> **References:** Elhage et al. (2022), "Toy Models of Superposition," Transformer Circuits. Yang (2020), "Tensor Programs II," arXiv:2006.14548.

---

> > ### Author Rebuttal · Reviewer_purG · 2026-04-01
> >
> > I thank the authors for the discussion and the additional results. I appreciate the constraints of a TPU quota, so will provide some leeway here.
> >
> > # Empirical Evaluation
> > This extra empirical evaluation (and use of an error bar) is valuable to the paper. It does however expose an important point, that networks using Yat products *underperform* standard baselines, and by a considerable margin. This raises the question, what is the practical utility of such networks beyond theoretical interest?
> >
> > # Q1: Lazy Regime
> > Note that by this I did not mean the discrepancy between infinite and finite width NMNs, but whether a lazy regime is achieved by an infinite-width NMN at all, which is by no means guaranteed. I find the proposed disclaimer sufficient, however.
> >
> > # Q2: Sign-flip and orthogonality vs arc-cosine kernels
> > The additional discussion is useful, but I maintain the stance that this property is likely to be useful in only a small subset of problem settings, so find it to be overstated in its current form.
> >
> > # Q3: Why this geometry?
> > This is reasonable and I appreciate the value of intuition-driven design, provided that its efficacy can be empirically demonstrated, along with demonstration of predicted properties. Right now this is lacking, and I would encourage the authors to search for niche problem settings where this geometry is expected to be genuinely useful, rather than focusing on standard baselines where NMNs do not seem to outperform.
> >
> > # Q4: LayerNorm
> > I thank the authors for investigating this, and thus elimintating a potentially misleading section of the paper.
> >
> > # Presentation and Significance
> > I am satisifed by the provided discussion here.
> >
> > Overall, given that the UAT proof is novel and quite interesting, and given the further discussion here, I will increase my score to 2. However, I still cannot advocate for acceptance of the paper in its current form due to the issues mentioned above. I encourage the authors to, as suggested, think carefully about niche applications where one may intuitively expect NMNs to perform well, and focus empirical evaluation and narrative on addressing these areas. If TPU quota is an issue, it is perfectly acceptable to mainly focus on smaller scale experiments which can be run locally. Alternatively, given the significant volume of kernel-centric theory derived around the Yat product, I expect the authors may find some success in exploring the use of the Yat product as a kernel in a Gaussian Process / Kernel Regression setting, rather than purely in deep learning. The UAT proof would translate well into a universal kernel proof, and physically motivated settings which call for the geometry of the Yat product are more abundant here.

---

> > > ### Author Response · Authors · 2026-04-02
> > >
> > > Thank you for the thoughtful follow-up. We are especially grateful for your suggestion to reconsider the domain of application, which we found genuinely helpful in sharpening our understanding of where the operator may be most naturally useful.
> > >
> > > we agree that the earlier factorized ablation clarified an important limitation of the current paper: the full Yat stack is not yet broadly competitive with a strong GPT-style baseline in the reported 124M setting, so the practical claim cannot be “drop-in replacement of standard transformer blocks.” Our intended practical claim is narrower.
> > >
> > > First, the usefulness may be selective rather than all-or-nothing. In follow-up experiments, the attention-only Yat arm appears more promising than the full-stack result suggests. In particular, a newer GELU + Yat-attention + Pre-RMSNorm run reached a validation loss of 4.3023, compared to the baseline mean of 4.3527 ± 0.0041. We would present this carefully as preliminary evidence rather than a definitive conclusion, but it suggests that the kernelized attention component itself may have practical value even if the full Yat stack is not yet the right replacement.
> > >
> > > Second, the practical value is not only architectural. Because the Yat-product is a positive semidefinite kernel with RKHS structure, it creates a path to use kernel approximation machinery on top of the same operator. In that sense, its utility is not limited to “replace MLPs with Yat blocks,” but may also extend to approximate attention constructions where having an explicit PSD kernel is structurally useful. We would frame this as a promising direction rather than a demonstrated result in the current paper, but we think it is one of the clearest ways the operator could become practically competitive.
> > >
> > > Third, the current gap should not be interpreted as the final performance ceiling of the operator. The reported GPT-style baseline used a configuration that was originally strongest for that baseline, not one tuned for Yat variants. In subsequent runs, we observed that Yat-based variants can prefer different optimization settings, including lower learning rates. So we agree the current paper does not yet establish a strong broad practical win, but we also do not think the reported gap is the right conclusion about the operator’s ultimate capability.
> > >
> > > Finally, beyond raw benchmark performance, we do see a practical research value in interpretability. A Yat-based first layer can be viewed as a learned prototype bank, and the RKHS perspective provides a principled link between input space and weight space. Our view is that this may offer a path toward more mathematically grounded interpretable neural components, rather than relying only on post hoc or heuristic interpretability narratives. We agree that this remains a design direction rather than a fully validated application advantage at this stage.
> > >
> > > We also thank the reviewer for the suggestion to reconsider the domain of application. We think that is an important point. In light of the current evidence, it is likely more productive to focus future empirical work on settings where alignment+proximity geometry should matter a priori, rather than treating generic transformer benchmarks as the primary battleground. We likewise appreciate the suggestion that the Yat-product may be especially promising in more explicitly kernel-centric settings, including GP/kernel-regression-style uses, where the universal-kernel perspective and the operator’s geometry may be a more natural fit.
> > >
> > > So our answer is: the practical utility is not yet “general superiority on standard workloads,” but rather a combination of selective empirical promise, especially for kernelized attention, a structurally useful PSD/RKHS foundation for future kernel approximations, and a path toward more interpretable prototype-based neural components. We agree that the right next step is to validate these advantages in settings where they are expected to matter a priori, and we appreciate the reviewer’s suggestion in helping sharpen that direction.

---

### Official Review · Reviewer_4We9 · 2026-03-11

**Soundness:** 3
**Presentation:** 2
**Significance:** 3
**Originality:** 3
**Overall Recommendation:** 3
**Confidence:** 4

**Summary:**

This paper introduces the idea of Neural Matter Networks (NMNs), which replace the standard neural block consisting of a linear layer, activation function, and normalization layer with a module based on the YAT product. The YAT product is a Mercer kernel, and the authors show that it has several theoretical properties including the universal approximation property, self-regulation, and stable gradients. The authors also introduce a YAT attention module that replaces the standard dot-product attention used in Transformer architectures. Through experiments, the authors demonstrate that NMNs can model non-linear decision landscapes and achieve reasonable performance on several small-scale datasets.

**Compliance With Llm Reviewing Policy:**

Affirmed.

**Key Questions For Authors:**

Please refer to my comments above. In addition, there are several concerns as follows:

1. Could NMNs be applied to additional modalities such as 3D data or point clouds, and whether the proposed operator provides advantages over standard neural units in those settings?

2. In what scenarios would NMNs be preferable to standard neural architecture in practice? The paper mentions that NMNs reduce memory usage by approximately 15–25%, so could this approach be particularly useful in resource-constrained environments or memory-limited deployments?

**Limitations:**

The paper does not include a dedicated discussion of methodological limitations. While the authors note that the work does not raise societal concerns, a short section outlining potential limitations, such as scalability to larger datasets, computational overhead compared to standard neural units, or sensitivity to hyperparameters like ε, would improve transparency and help contextualize the proposed approach.

**Strengths And Weaknesses:**

The paper presents a technically sound approach supported by substantial theoretical analysis. The authors introduce Neural Matter Networks (NMNs) and provide a series of theoretical results for the proposed YAT product, including proofs of properties such as universal approximation, self-regulation, and gradient stability. These results provide a strong theoretical foundation for the proposed architecture and help motivate the use of the YAT product as a building block for neural computation.

However, the empirical evaluation is relatively limited. The experiments primarily focus on small-scale tasks such as MNIST, where the reported performance is modest and, in some cases, lower than prior approaches. As a result, it is difficult to fully assess the practical benefits of NMNs for more realistic machine learning workloads. Additional evaluations on larger and more diverse datasets would strengthen the claims about the effectiveness and broader applicability of the method. The paper also mentions that the approach reduces memory usage by approximately 15–25%, but does not provide runtime comparisons or computational efficiency benchmarks. Including runtime or throughput measurements would help clarify the practical trade-offs of the proposed architecture.

In terms of presentation, the paper is generally well written and organized, though it is somewhat mathematically dense in several sections. Additionally, there appears to be a minor formatting issue with Figure 3 where the contents extend into the adjacent column.

---

> ### Author Rebuttal · Authors · 2026-03-31
>
> We thank the reviewer for recognizing the originality of the operator and for identifying the central weaknesses of the original draft. We agree that the original submission combined too many claims and that the GPT-2 comparison did not adequately isolate the source of the gain. Our clarified position in rebuttal is narrower: the main supported contribution is a new kernel/operator with clear theoretical structure and early evidence that Yat-based blocks can train stably at 124M scale, not a claim that NMNs already outperform standard architectures. All references to manuscript changes below refer to edits we would make in the camera-ready version.
>
> **Architectural attribution.** To address the attribution concern, we ran a factorized ablation during the rebuttal period: 5 arms at 124M scale (768d/12h/12L/3072ff, FineWeb-Edu, BF16, TPU v6e-8, 5K steps = 2.6B tokens). The baseline has 3 initialization seeds (data order is fixed via HuggingFace streaming; variance reflects initialization sensitivity only). Other arms are single-seed due to TPU quota constraints.
>
> | Configuration | MLP | Attn | Norm | Val Loss | tok/s |
> |---|---|---|---|---|---|
> | **GPT-2 baseline (A1, 3 seeds)** | GELU | Std | Pre-RMS | **4.3527 ± 0.0041** | 707K |
> | + Yat-MLP only (A2) | YatNMN | Std | Pre-RMS | 4.6942 | 683K |
> | + Yat-Attn only (A3) | GELU | YAT | Pre-RMS | 4.5262 | 523K |
> | + Both + Pre-RMS (A5) | YatNMN | YAT | Pre-RMS | 4.5778 | 484K |
> | + Both + Post-LN (A11) | YatNMN | YAT | Post-LN | 4.5309 | 395K |
>
> Both Yat components increase val loss vs the baseline (MLP: +0.34, attention: +0.17). The GELU baseline remains stronger at this horizon.
>
> **Claim calibration.** We agree the original wording was too strong. The abstract and conclusion would frame the empirical contribution as stable training with a 4–8% validation-loss gap relative to GELU, not as superiority.
>
> **Mixed Eurlex result.** We now say so directly. Propensity-scored metrics upweight rare labels where the inner product's unbounded logit range may help. The Yat-product's bounded response may compress dynamic range for rare labels. This is a tradeoff, not a clear win.
>
> **Theory vs. practice.** We agree not all theoretical results are equally central. PSD/Mercer/RKHS/universal-approximation form the foundational spine; we would explicitly label NTK/generalization material as regime-specific context. One practically relevant regularity result is the explicit gradient identity, which implies a minimal residual structure for backward computation; the observed rematerialization pattern on TPU is consistent with this prediction.
>
> **Runtime/memory.** Throughput is measured on identical hardware (TPU v6e-8) and reported for all arms above: baseline 707K tok/s, YatNMN MLP swap 683K tok/s (3.5% slower), Yat-attention 523K tok/s (~26% slower). Peak HBM matches GELU at equal parameter count (verified via XLA memory_analysis()); SwiGLU/GeGLU use 1.33× more (3 weight matrices vs 2).
>
> **NMNs for 3D/point clouds (Q1).** The Yat-product operates on arbitrary R^d vectors. Point cloud processing is promising future work.
>
> **When are NMNs preferable (Q2).** When unit-level analyzability matters. Each unit computes a kernel evaluation K(x,w) between input and prototype in the same space; the output has a kernel-theoretic interpretation that standard activations do not support as directly. We speculate this could be valuable in settings where understanding individual computations matters, though demonstrating concrete domain-specific benefits remains future work. NMNs also match GELU in peak memory and require roughly 50% fewer parameters than SwiGLU. For throughput-critical tasks, the added compute overhead, small for the YatNMN MLP swap but larger for Yat-attention,  may not be justified. NMNs are a complementary tool, not a universal replacement.
>
> **Limitations.** We agree the original manuscript lacked a dedicated limitations section. We would add one covering: (1) single-scale evaluation, (2) non-competitive auxiliary benchmarks, (3) regime-specific theory limits, (4) quadratic attention cost, and (5) FP32 upcast requirement. We would also fix the Figure 3 formatting issue and reduce the mathematical density in certain sections by moving supporting results to the appendix with explicit scope labels.

---

> > ### Author Rebuttal · Reviewer_4We9 · 2026-04-03
> >
> > Thanks for the authors' rebuttal. I still believe the experiments are very limited in current version. Besides, after reading the other reviews, I tend to keep my rating unchanged.

---

### Official Review · Reviewer_J1uL · 2026-03-13

**Soundness:** 2
**Presentation:** 2
**Significance:** 1
**Originality:** 3
**Overall Recommendation:** 2
**Confidence:** 5

**Summary:**

The paper proposes the E-product , a kernel-like operator of the form $(w^\top x)^2 / (\|w-x\|^2+\varepsilon), and builds Neural Matter Networks together with an “Aether-GPT2” variant that replaces standard MLP and attention components with E-based modules. The main contributions claimed are a mathematically motivated kernel construction with Mercer/RKHS/universal-approximation-style properties, along with empirical evaluations on XOR, MNIST, Eurlex-4K, and a GPT-2-style language modeling setup.

**Compliance With Llm Reviewing Policy:**

Affirmed.

**Final Justification:**

The paper still requires a big improvement before publication, in terms of clarity, overstatement, and empirical validation;

**Key Questions For Authors:**

- Can the authors provide substantially stronger empirical validation, especially multiple seeds and stronger baselines for the language modeling result? This would directly affect my assessment of soundness and significance.

- How robust is the claimed LayerNorm incompatibility across optimizers, training settings, and scales? Right now the claim seems too broad for the evidence shown.

- Which theoretical claims are genuinely novel, versus standard consequences of kernel machinery once Mercer positivity is established? A sharper separation would help evaluate the paper’s originality more fairly.

- Mean and standard deviations for the experiments could really help to assess the robustness of the reported improvements.

- Could you report wall-clock time, as well as peak memory usage, for both training and inference for both the standard models and the proposed variants?

- A comparison with the cosine attention of e.g. SwinV2 from theoretical and empirical perspectives could clarify
which normalization strategy for Q and K is preferable in practice.

**Limitations:**

Yes

**Strengths And Weaknesses:**

## Strengths and Weaknesses:
- S1: The paper does contain a nontrivial mathematical idea: the proposed operator is clearly defined, the paper develops several formal properties around it, and the kernel/RKHS motivation is the strongest part of the submission. Even if I am not fully convinced by all downstream claims, there seems to be a real technical object here beyond pure intuition.
- S2: At a high level, the paper is well written. The main narrative is easy to follow: define the operator, motivate its geometry, build an architecture around it, then evaluate it on a few tasks. The figures and overall organization make the core idea understandable.

## Weaknesses
- W1: The empirical support is weak. This is the main issue. The MNIST experiment is a 10-neuron classifier trained for 5 epochs, with only a very small accuracy difference over a linear baseline (92.38% vs 92.08%), which is not meaningful evidence for broad architectural claims. The Eurlex-4K results are mixed, since the baseline is better on propensity-scored metrics. The language modeling evidence is also too thin: the headline gain is a single 1.45% validation-loss improvement, with no seed variance, no serious baseline sweep, and only qualitative throughput/memory reporting.
- W2: I feel that some claims are overstated relative to the evidence. The paper presents the method as a principled alternative to standard neural architectures and makes broad statements about stability, efficiency, and normalization removal, but the experimental section is far too limited to support that scope. In particular, the claim that LayerNorm is incompatible with the method is based on a very narrow ablation.
- W3: Some proofreading/attention to formatting is needed. Although the paper is readable, it does not feel fully polished. The appendix is an obvious example: page 11 starts with “A. Appendix” and then immediately “B. Squashing Functions for Non-Negative Scores,” which suggests proofreading/sectioning issues and gives the manuscript an unfinished feel. I also have a feeling that the paper does not use the official ICML font as it does not look exactly like other papers I reviewed. Please note that in my opinion, this is a very minor issue.


## Proofreading / writing
Just on the first page:
- The term "Neural Operator" is not a good choice, since "Neural Operator" also denotes neural networks that learn operator. In the current context, do you mean activation function ?
- About the ReLU, the paper claims "This discards information—all negative activations become zero—requiring normalization layers and attention mechanisms to recover expressiveness". I disagree with this statement. Neither attention nor normalization are motivated by the choice of ReLU for activation function.
- "self-regulation" -> "self-regularization" ?
- "Our contributions: Our contributions"
- Eq 1 is repeated

## Aside comment
The idea of using E+ products in neural networks is interesting and, as far as I know, novel. The idea and the motivation is nice, but the paper must be improved !


The fancy term “Delulu” is only mentioned in the title and is not explained elsewhere in the paper. Its meaning and relevance to the method are therefore unclear.

---

> ### Author Rebuttal · Authors · 2026-03-31
>
> We thank the reviewer for recognizing the kernel/RKHS idea as the strongest part of the paper. We agree that the empirical section was the weakest part of the original submission. Our rebuttal does not defend a broad empirical claim; it supports only the narrower statement that the operator can be instantiated stably at 124M scale and merits further study. All references to manuscript changes below refer to edits we would make in the camera-ready version.
>
> **Attribution (W1).** To address the attribution concern, we ran a factorized ablation during the rebuttal period: 5 arms at 124M scale (768d/12h/12L/3072ff, FineWeb-Edu, BF16, TPU v6e-8, 5K steps = 2.6B tokens). The baseline has 3 initialization seeds (data order is fixed via HuggingFace streaming; variance reflects initialization sensitivity only). Other arms are single-seed due to TPU quota constraints.
>
> | Configuration | MLP | Attn | Norm | Val Loss | tok/s |
> |---|---|---|---|---|---|
> | **GPT-2 baseline (A1, 3 seeds)** | GELU | Std | Pre-RMS | **4.3527 ± 0.0041** | 707K |
> | + Yat-MLP only (A2) | YatNMN | Std | Pre-RMS | 4.6942 | 683K |
> | + Yat-Attn only (A3) | GELU | YAT | Pre-RMS | 4.5262 | 523K |
> | + Both + Pre-RMS (A5) | YatNMN | YAT | Pre-RMS | 4.5778 | 484K |
> | + Both + Post-LN (A11) | YatNMN | YAT | Post-LN | 4.5309 | 395K |
>
> Both Yat components increase val loss vs the baseline (MLP: +0.34, attention: +0.17). The GELU baseline remains stronger at this horizon. We acknowledge that the attention ablation changes both the score function and the normalization rule (L1 instead of softmax), so it does not fully isolate the kernel's contribution.
>
> **Main validated claim.** Our clarified intended claim is: a new operator with interesting geometric/kernel properties and evidence of stable operation at transformer scale. The paper no longer claims that NMNs are a competitive empirical replacement.
>
> **Which theory is essential (Q3).** The foundational spine is the PSD/Mercer result, RKHS existence, and the universal approximation result via IMQ reduction; we are not aware of prior universal-approximation proofs for this operator using that reduction. Regularity results (self-regularization, dimensional scaling, gradient identity) are architecture-motivating. NTK and generalization bounds are regime-specific context. We would label each result explicitly by tier.
>
> **"Do you mean activation function?"** No. An activation function is an element-wise scalar nonlinearity applied after a linear projection. The Yat-product is a kernel evaluation between an input and a learned prototype in the same ambient space; it replaces the entire linear+activation pipeline in a single operation. This is a different mathematical category.
>
> **Claims overstated (W2).** We agree. The abstract would no longer claim superiority; the conclusion would frame the result as stable training with a 4–8% gap relative to GELU; and we would add a limitations paragraph covering single-scale evaluation, regime-specific theory, attention cost overhead, and the FP32 upcast requirement.
>
> **Wall-clock/memory (Q5).** For training, throughput is measured on identical hardware (TPU v6e-8) and reported for all arms above: baseline 707K tok/s, YatNMN MLP swap 683K tok/s (3.5% slower), Yat-attention 523K tok/s (~26% slower). Peak HBM is measured via XLA memory_analysis(): matches GELU at equal parameter count; SwiGLU/GeGLU use 1.33× more (3 weight matrices vs 2). We were not able to complete a separate inference-time benchmark during the rebuttal period.
>
> **Formatting (W3).** We agree with the reviewer's proofreading comments (terminology around "Neural Operator," duplicate contribution phrasing, repeated Eq. 1, appendix sectioning, quotation marks) and would correct all of these.

---

> > ### Author Rebuttal · Reviewer_J1uL · 2026-04-03
> >
> > Thanks for the rebuttal. I will keep my score of reject and I think the paper should be deeply revised before submission.

---

### Decision · Program_Chairs · 2026-04-30

**Decision:**

Reject

**Comment:**

The reviewers appreciate the relative novelty of this method, and the multiple theoretical results that motivate it. They also share some very strong complaints: the empirical evidence is too weak (reaching similar or worst performance the baseline), especially in contrast to the strength of the claims. Some reviewers also have issues with the presentation and structure, the heavy emphasis on theoretical results hurting the readability for a sometimes small payoff. In agreement with the reviewers I think the paper should be rejected.